# How Low Can You Go? Identifying Prototypical In-Distribution Samples for Unsupervised Anomaly Detection

## Abstract

Unsupervised anomaly detection (UAD) alleviates large labeling efforts by training exclusively on unlabeled in-distribution data, through which outliers (out-of-distribution data) are detected as anomalies. While generally the assumption prevails that larger training datasets improve UAD performance, we find that training UAD models with extremely few carefully selected samples can match—or even surpass—the performance of training on the entire dataset. To investigate this effect, we introduce an unsupervised method to identify a compact core-set of prototypical samples boosting UAD performance, when training with only a select few samples. Our analysis across seven diverse UAD benchmarks from computer vision, industrial defect detection, and medicine shows that with just 25 selected samples, we exceed full-dataset training performance in 25 out of 67 categories. Furthermore, we find that the selected core-set of prototypical samples generalizes well across models and datasets, providing important insights into their in-distribution nature. These samples exhibit clear, unobstructed, high-fidelity characteristics, which highlights the importance of data quality over quantity in UAD training. The code is available at https://anonymous.4open.science/r/uad_prototypical_samples/

## 1 Introduction

Unsupervised anomaly detection (UAD), also known as out-of-distribution (OOD) detection, aims to distinguish in-distribution (ID) samples from those originating from a different distribution. To achieve this, machine learning models are commonly trained on exclusively ID data to learn its underlying characteristics. Once trained, the model detects OOD samples by assessing their distance from the learned distribution. Compared to supervised training, this setup alleviates the need for large labeled datasets, is not susceptible to class imbalance, and is not restricted to anomalies seen during training. Due to these advantages, UAD has several vital applications in computer vision: It is used to detect pathological samples in medical images (Schlegl et al., 2019; Lagogiannis et al., 2023; Bercea et al., 2022; Meissen et al., 2023), to spot defects in industrial manufacturing (Bergmann et al., 2019; Roth et al., 2022; Deng & Li, 2022; Bae et al., 2023), or as safeguards to filter unsuitable input data for supervised downstream models, for example, in autonomous driving.

In deep learning, the prevailing assumption is that more data leads to better models. However, suppose it were possible to train models using only a small fraction of the commonly used data while still achieving strong performance—then, this would offer numerous advantages, as large amounts of data, especially labeled data, are often difficult and costly to obtain. The medical sector is particularly good example for this. Small datasets on the contrary, are cheap, easy to collect, and accessible for a wider range of applications. This would make problems where data is difficult or expensive to acquire much more feasible to tackle. Moreover, UAD models are especially suited for training with extremely small datasets, because, compared to supervised models—where overfitting on a few training samples diminishes their utility as generalization to other classes is inhibited—these models are not impacted in the same way. In fact, they rely on overfitting to the ID data to detect OOD data. Furthermore, small training datasets lead to better explainability, since output scores can directly be related to (dis)similarities in the training data. To summarize, the ability to train UAD models with smaller datasets would not only lower the entry barrier for designing high-performing

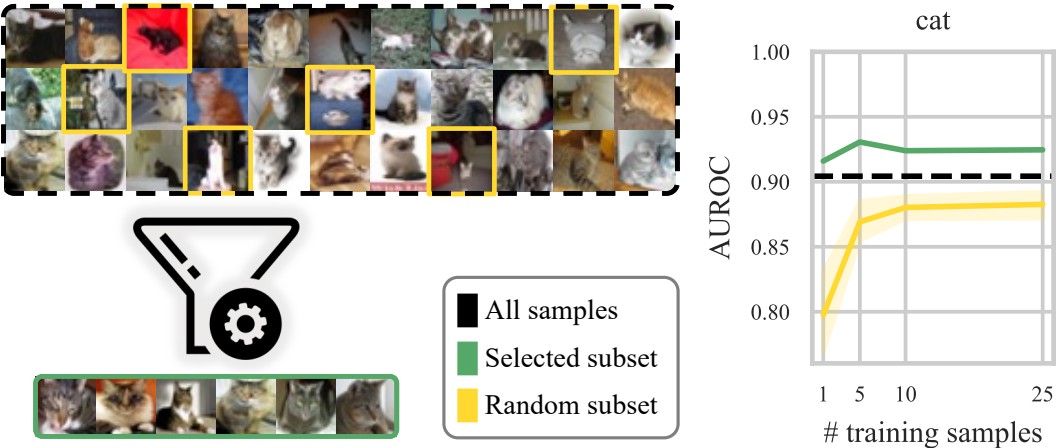

Figure 1: Using our method to select only a few prototypical in-distribution samples for training can result in higher UAD performance compared to training with 100% of the available data. Results for the *cat* class from CIFAR10. Black dashed: full training. Yellow: randomly selected samples, including standard deviations over different random selections. Green: Best-performing samples identified with our method.

algorithms—especially for actors with limited resources, thus contributing to the democratization of AI—but would also make the training of such models more resource-efficient.

Motivated by the observation that smaller, well-chosen subsets of the training data can, in certain cases, yield surprisingly strong performance, this paper further explores this phenomenon in the context of UAD. Particularly, through the unification of concepts from UAD and core-set selection, we propose an unsupervised method for selecting a high-performing subset of samples from the initial training dataset, as depicted in Figure 1, and evaluate the effectiveness of this approach on a multitude of different models, datasets, and tasks. To further strengthen our results, we also utilize two weakly supervised sample selection strategies as weak upper bounds for the introduced unsupervised sample selection and provide additional insights into the method's selection process.

Our findings suggest that training a UAD model on a carefully selected subset of very few ($\leq 25$) training samples can indeed achieve performance comparable to—or in some cases even exceeding—that of models trained on the entire available dataset (which we denote as "full training" in the remainder of the manuscript), and that the selected prototypical samples underscore the importance of data quality over quantity. In summary, the main contributions of this paper are:

- We show for the first time that an exceedingly small number of training samples can suffice for performant, robust, and interpretable UAD, achieving state-of-the-art performance on a multitude of established benchmarks (Section 5.1).

- We propose an unsupervised method (Section 3) to reliably find well-performing subsets of prototypical in-distribution samples and describe their common characteristics (Section 5.2).

- We further demonstrate that the prototypical samples identified by our method, along with their characteristics, generalize well across different models, datasets, and even tasks, yielding consistently strong performance (Section 5.3).

## 2 Related Work

Our work combines ideas from both the fields of Unsupervised Anomaly Detection (UAD) and core-set selection. Here, we give a brief overview of these concepts and related research work.

## 2.1 Unsupervised Anomaly Detection (UAD)

UAD is deeply rooted in many domains, including computer vision, with many influential works benchmarking their models on natural-image datasets, such as CIFAR10 and CIFAR100 (Krizhevsky et al., 2009), MNIST (LeCun et al., 1998), or Fashion-MNIST (Xiao et al., 2017). Early works attempting to solve UAD on these benchmarks were mostly based on (variational) autoencoders (Zhou & Paffenroth, 2017; Kim et al., 2019; Liu et al., 2020; Abati et al., 2019) or GANs (Perera et al., 2019; Deecke et al., 2019; Akcay et al., 2019) trying to restrict the learned manifold of the generative model. The models are then expected to faithfully reconstruct in-distribution samples, through which OOD samples can be detected due to their large reconstruction errors. Also, one-class classification models (Ruff et al., 2018) or ones that learn surrogate tasks (Golan & El-Yaniv, 2018; Bergman & Hoshen, 2020) have been successfully used. More recently, works based on pre-trained neural networks (such as ResNets He et al. (2016)) have become popular and still are the best-performing models for the aforementioned datasets (Bergman et al., 2020).

The release of MVTec-AD (Bergmann et al., 2019) for industrial defect detection sparked several works focusing on this dataset as it was the first to contain a variety of useful, real-world anomaly detection tasks. After early attempts to solve these with various techniques, including autoencoders (Bergmann et al., 2018) and knowledge distillation methods (Bergmann et al., 2020), research converged on self-supervised approaches (Zavrtanik et al., 2021; Li et al., 2021), ResNets pre-trained on ImageNet (Defard et al., 2021; Deng & Li, 2022; Roth et al., 2022), or combinations thereof (Bae et al., 2023).

Anomaly detection was also successfully applied in medical computer vision, where it is used to discriminate samples from healthy subjects (in-distribution) from diseased ones (out-of-distribution). Schlegl et al. (2019) have successfully discovered biomarkers in retinal OCT images using a GAN. To detect tumors and lesions in brain MRI, numerous autoencoder-based approaches (You et al., 2019; Baur et al., 2021; Zimmerer et al., 2019) and diffusion models (Wyatt et al., 2022) have been proposed. Furthermore, anomaly detection has been successfully applied in chest X-ray images to detect COVID-19 (Zhang et al., 2020) or other malignancies (Lagogiannis et al., 2023; Mao et al., 2020).

However, without exception, these existing works have followed the established paradigm of using the largest available training dataset, a notion we aim to challenge in this study.

## 2.2 Core-set Selection

To this end, we utilize methods from the field of core-set selection. Core-set selection aims to create a small informative dataset such that the models trained on it show a similar test performance compared to those trained on the full dataset. Core-set selection techniques for deep learning include minimizing the feature-space distance (Welling, 2009) or the distance of gradients with respect to a neural network (Mirzasoleiman et al., 2020) between the selected subset and the full training dataset. In anomaly detection, core-set selection has been used by Roth et al. (2022). Here, however, the selection is done on patch features instead of images. In MemAE, Gong et al. (2019) restricted the latent space of an autoencoder to a set of learned in-distribution feature vectors to perform anomaly detection. While this work also finds prototypical feature vectors, they again cannot be linked back to training samples and, consequently, cannot be used for core-set selection.

## 3 Surfacing Prototypical In-Distribution Samples Through Core-Set Selection

In UAD, the models objective is to learn a decision function $f : X \to \{-1, 1\}$ that assigns a binary label to each sample $x \in X$, where $X \subset \mathbb{R}^D$ corresponds to the set of all possible samples. Specifically, we define the label $y(x) = -1$ if $x$ is considered an in-distribution (ID) sample, and $y(x) = 1$ if $x$ is deemed an out-of-distribution (OOD) sample. In the typical unsupervised setting, the training dataset $X_{\text{train}} \subset X$ is assumed to contain only ID samples. That is, for every $x \in X_{\text{train}}$, the true label satisfies $y(x) = -1$. This assumption eliminates the need for extensive labeling of OOD samples for training process, requiring only a smaller labeled validation dataset.

Without loss of generality, the UAD models introduced in Section 2, as well as the ones used in this work, can be divided into two parts: The feature extractor $\psi : X \to Z$ maps a sample $x \in X$ to its latent representation

$z \in Z$, while the predictor $\phi : Z \to \mathbb{R}$ computes an anomaly score $s \in \mathbb{R}$ from $z$. The anomaly score $s$ is a potentially unbounded continuous value that represents the "outlierness" of a sample $x$. Combined, the UAD model $\theta$ computes the anomaly score of a sample as:

$$\theta(x) = \phi(\psi(x)) = s \, .$$

These models are usually trained on large datasets. Let $X_{\text{train}} = \{x_i \in X \mid y(x_i) = -1, \ i = 1, 2, \dots, N\}$ denote the full training dataset, where $N \in \mathbb{N}$ represents the number of samples in the training set and $y(x_i)$ is the label associated with $x_i$. However, in core-set selection, our goal is to determine a subset $X_{\text{sub}} \subset X_{\text{train}}$ with $|X_{\text{sub}}| = M$ samples, where $M \in \mathbb{N}$ and $M < N$. As such we want to

$$\begin{aligned} \text{minimize} \quad & E(X_{\text{sub}}, \theta) \quad \text{subject to} \\ & X_{\text{sub}} \subset X_{\text{train}}, \quad |X_{\text{sub}}| = M, \quad M < N, \end{aligned} \tag{1}$$

where $E(X_{\text{sub}}, \theta)$ is a detection error produced by a model $\theta$ trained on $X_{\text{sub}}$.

Previous work by Defard et al. (2021) has shown that the latent space $Z_{\text{train}} = \{\psi(x) \in Z \mid x \in X_{\text{train}}\}$ represents a semantically meaningful compression of the training distribution $X_{\text{train}}$ that can be modeled as a multivariate Gaussian. Due to the limited representational power of unimodal Gaussian distributions, we extend this idea by instead fitting a Gaussian Mixture Model (GMM) with $K$ components to this latent space. Let $\mathcal{G} = \{\mu_1, \mu_2, \dots, \mu_K\}$ represent the set of means (centroids) of the $K$ components of the GMM, where $\mu_j$ denotes the mean of the $j$-th component. This allows us to choose the samples corresponding to the latent codes that are closest to the centroids as core-set samples.

$$X_{\text{sub}} = \left\{ x_i \mid \forall \mu_j \in \mathcal{G}, \underset{x_i \in X_{\text{train}}}{\arg \min} \, ||\psi(x_i) - \mu_j||_2 \right\} . \tag{2}$$

The use of a GMM ensures that multiple different modes of normality are represented in $X_{\text{sub}}$. For the implementation details of this method please refer to the code.

### 3.1 Weakly-Supervised Baselines

In addition to our unsupervised core-set selection, we are interested in identifying the optimal training subsets $X_{sub,M}^*$ of size $M$ when labeled data is available. For example, the optimal training subset for $M = 5$ would consist of the five samples from $X_{train}$ that together minimize the model's detection error. Finding these subsets would allow us to benchmark our previously introduced approach. In principle, one could determine the optimal subset of training samples—those that minimize the detection error—by exhaustively evaluating all possible subsets of size $M$ from a dataset of size $N$. However, this is computationally infeasible due to the combinatorial explosion of possibilities. Specifically, there are $\binom{N}{M}$ possible subsets, making exhaustive search impractical even for moderate values of $M$ and $N$. To this end, we propose two approximate, weakly supervised, strategies—a greedy algorithm (Section 3.1.1) and an evolutionary algorithm (Section 3.1.2)—which identify small, high-performing training subsets. Finally, while there is no guarantee that the optimal training subset is found, we want to emphasize that these weakly supervised baselines are included solely as benchmarks for our core-set selection method introduced in Section 3.

### 3.1.1 Greedy Selection

Since even training on a fraction of the $\binom{N}{M}$ possible subsets $X_{sub,M}$ in a greedy scenario remains computationally expensive, we first heuristically estimate the quality of each $x \in X_{\text{train}}$ individually. To achieve this, we train our model $\theta$ on each sample $x_i$ for $i = 1, 2, \dots, N$ separately and assess its quality using $E(\{x_i\}, \theta, X_{\text{val}})$, where we define $E$ as the AUROC score, our chosen optimization target. The AUROC, or *Area under the Receiver Operating Characteristic Curve*, is a well-estasblished metric defined by the trade-off between the True Positive Rate (TPR) and the False Positive Rate (FPR), where TPR measures the proportion of actual positives correctly identified, and FPR measures the proportion of actual negatives incorrectly classified as positives (Nahm (2022)). Notably, this approach is equivalent to identifying the best training subset $X_{\text{sub},1}^*$, by simply trying all $N$ subsets of size $M = 1$.

With this in mind, we construct $X_{\text{sub}}$ as:

$$X_{\text{sub}} = \underset{\substack{S \subseteq X_{\text{train}} \\ |S|=M}}{\arg\min} \sum_{x \in S} E(\{x\}, \theta, X_{\text{val}}) \,. \tag{3}$$

Note that the set of individual samples that produce the smallest errors is not necessarily the same as the subset that minimizes the overall error when trained together. This is because membership in $X_{\text{sub}}$ is determined based on the performance of $\theta$ trained on each sample $x$ in isolation, rather than on the joint performance of $\theta$ trained on all samples in $X_{\text{sub}}$ simultaneously.

### 3.1.2 Evolutionary Algorithm

While the greedy approach above is intuitive, fast, and easy to implement, it prefers subsets of visually similar samples, as shown in Section 5.6. While this is desirable in some cases, there are also scenarios in which multiple modes of normality should be covered by the selected subset. As such, to get a better coverage of the normal variations in a dataset, we propose a second approach to finding good subsets $X_{\text{sub},M}$.

For each combination of a training sample $x_i \in X_{\text{train}}$ and a validation sample $x_k \in X_{\text{val}}$, we compute an anomaly score $s(x_i, x_k)$ by training the UAD model $\theta$ on $x_i$ only and running inference on $x_k$. The objective is to find a subset $X_{\text{sub},M} \subset X_{\text{train}}$ that maximizes a fitness function $f$ described as:

$$f(X_{\text{sub},M}) = \sum_{(x_k, y_k) \in X_{\text{val}}} \max_{x_i \in X_{\text{sub}}} \left( y_k \cdot s(x_i, x_k) \right) \,. \tag{4}$$

Maximizing $f$ allows for finding $M$ training samples $x_i$ that achieve the best performance in classifying validation samples $x_k$ as ID or OOD. Given the vast number of possible subsets, finding an optimal solution directly is computationally infeasible. To approximate a high-quality solution, we employ an evolutionary algorithm:

---

**Algorithm 1:** Evolutionary Algorithm

**Data:** Training dataset $X_{\text{train}}$, validation dataset $X_{\text{val}}$, population size $P$, anomaly scores
$s(x_i, x_k) \,\forall x_i \in X_{\text{train}}, x_k \in X_{\text{val}}$, fitness function $f$, number of generations $G$
**Result:** Approximately optimal subset $\hat{X}_{\text{sub}}$
Initialize a random population $\mathcal{P} = \{X_{\text{sub}}^{(p)} \mid 1 \le p \le P\}$, where each $X_{\text{sub}}^{(p)} \subset X_{\text{train}}$ is a randomly
sampled subset of size $M$, i.e., $|X_{\text{sub}}^{(p)}| = M$.
**for** $gen \leftarrow 1$ **to** $G$ **do**

    Evaluate the fitness function $f$ for each individual $X_{\text{sub}}^{(p)} \in \mathcal{P}$;

    Remove the least-fit $\frac{P}{2}$ individuals $X_{\text{sub}}^{(p)} \in \mathcal{P}$ from $\mathcal{P}$ to determine the best subset $\mathcal{P}'$;
    Randomly apply either a crossover (combine two individuals) or mutation (replace one sample)
     operation to each individual in $\mathcal{P}'$ to create a modified population $\mathcal{P}''$;
    Generate a new population $\mathcal{P} = \mathcal{P}' \cup \mathcal{P}''$;

**end**
**return** *Best individual found in the final population*;

---

In the crossover operation, random subsets of two individuals $X_{\text{sub}}^{(1)}$, $X_{\text{sub}}^{(2)} \in \mathcal{P}'$ (called parents) are merged to produce a new individual $X'_{\text{sub}}$, such that $|X'_{\text{sub}}| = M$. The merging is performed using a random binary mask $\mathbf{b} \in \{0,1\}^M$ where each element $b_i$ is sampled independently from a Bernoulli distribution with $p = 0.5$. The offspring $X'_{\text{sub}}$ is then constructed element-wise as:

$$X'_{\text{sub},i} = \begin{cases} X_{\text{sub},i}^{(1)} & \text{if } b_i = 1 \\ X_{\text{sub},i}^{(2)} & \text{if } b_i = 0 \end{cases}$$

where $X'_{\text{sub},i}$, $X_{\text{sub},i}^{(1)}$, and $X_{\text{sub},i}^{(2)}$ denote the $i$-th elements of the offspring and parent individuals respectively. In the mutation operation, one sample $x_1 \in X_{\text{sub}}$ of an individual $X_{\text{sub}} \in \mathcal{P}'$ is randomly replaced with another sample $x_2 \in X_{train} \setminus X_{\text{sub}}$ to produce a new individual $X'_{\text{sub}} = X_{\text{sub}} \setminus \{x_1\} \cup \{x_2\}$.

In contrast to the greedy selection strategy that favors visually similar samples, the subsets found by the evolutionary algorithm have better coverage of the different notions of normality contained in the training dataset (c.f. Figure 8). However, note that this enhanced coverage could also be harmful when the normal dataset is noisy and contains samples that should be considered abnormal. In such a scenario, the greedy approach is more effective at filtering out these samples.

## 4 Experiments

### 4.1 Datasets and Models

To evaluate our methods, we use datasets from the natural- and medical-image domains, showing the applicability of our method in diverse tasks. CIFAR10, CIFAR100 (Krizhevsky et al., 2009), MNIST (LeCun et al., 1998), and Fashion-MNIST (Xiao et al., 2017) are trained in a one-vs-rest setting, where one class is used as the in-distribution, and all other classes are combined as outliers. MVTec-AD (Bergmann et al., 2019) is an industrial defect detection dataset and a frequently used benchmark for UAD models. In the chest X-ray images of the RSNA Pneumonia Detection dataset (Stein et al., 2018), the in-distribution constitutes images of healthy patients, while anomalous samples show signs of pneumonia or other lung opacities. In addition, we use CheXpert (Irvin et al., 2019) to test if the samples found by our method generalize to other datasets. Similarly to RSNA, the in-distribution samples here are images labeled with "No Finding", while OOD samples either display pneumonia or other lung opacities. Lastly, we detect MRI slices with glioma in the BraTS dataset (Menze et al., 2014; Bakas et al., 2017). For each dataset, we chose a respective state-of-the-art model: PANDA (Reiss et al., 2021) is used for CIFAR10, CIFAR100, MNIST, and Fashion-MNIST, PatchCore (Roth et al., 2022) for MVTec-AD, and FAE (Meissen et al., 2023; Lagogiannis et al., 2023) for RSNA, CheXpert and BraTS. We additionally used Reverse Distillation (RD) by Deng & Li (2022) for RSNA to test the generalizability of identified samples across models. Details about the datasets and models can be found in the supplementary material (see Appendix B).

### 4.2 Experimental Setup

We train all models using the original hyperparameters from their respective literature unless explicitly stated otherwise and also use the official PyTorch implementations provided by authors. PANDA is trained for a constant 2355 steps (equivalent to 15 epochs on CIFAR-10) following the recommendations of Reiss et al. (2021), using the SGD optimizer with a learning rate of 0.01 and weight decay of 0.00005. FAE, which converges quickly on BraTS, RSNA, and CheXpert, is trained for 500 steps with the Adam optimizer using a learning rate of 0.0002.

For our unsupervised core-set selection, we employ a Gaussian Mixture Model (GMM) to identify representative training samples. The GMM is fitted to the feature space extracted from the model, with the number of components set to the desired subset size $M$. The model is initialized with a fixed random seed, and scikit-learn's default parameters are used for fitting. Our greedy and evolutionary weakly supervised baseline algorithms only require few parameters. The former requires only a single parameter: the subset size $M$. The latter operates with a fixed population size of $P = 1000$ and runs for $G = 500$ generations, in addition to the subset size $M$. Our experiments indicate that the evolutionary approach remains robust across variations in these parameters. Finally, note that we set the maximum subset size to $M = 25$ samples for all three algorithms, as no significant performance gains were observed beyond this threshold. This choice ensures a balance between computational efficiency and comprehensive evaluation across different selection strategies.

For further details, we kindly refer the reader to the codebase (see Abstract), which includes, but is not limited to the full dataset configurations/splits, training procedures, models, our unsupervised core-set selection strategy, our weakly supervised baselines, and the exact experiment settings.

Table 1: AUROC scores on the test set for $M \in \{|X_{\text{train}}|, 1, 5, 10, 25\}$ samples, selected using our unsupervised core-set selection method (S. 3), as well as the weakly supervised greedy (S. 3.1.1) and evolutionary (S. 3.1.2) algorithms. "Random" refers to a baseline where subsets are sampled uniformly without replacement from $X_{train}$. The $\pm$ values for random selection indicate the standard deviation across ten different runs. Best overall performances are marked in **bold**, and underlined numbers indicate the best performance for each sample size. Numbers marked with * surpass full training with $X_{train}$. This table reports the averaged results across all classes in each dataset; detailed per-class results are in Appendix C.

| | Method | CIFAR10 | CIFAR100 | F-MNIST | MNIST | MVTec-AD | BraTS | RSNA |
|---|---|---|---|---|---|---|---|---|
| **1 sample** | Random | 84.96 ±1.2 | 77.05 ±1.4 | 83.51 ±3.3 | 76.11 ±2.2 | 83.61 ±1.2 | 93.85 ±5.3 | 61.86 ±5.2 |
| | Greedy | 95.02 | 89.94 | 94.00 | 90.15 | 89.70 | 95.25 | 70.64 |
| | Evo | 87.68 | 79.99 | 86.82 | 74.82 | 85.55 | 94.00 | 65.62 |
| | Core-set | 90.08 | 79.62 | 91.47 | 82.29 | 82.80 | 96.25 | 67.36 |
| **5 samples** | Random | 91.95 ±0.7 | 86.40 ±0.8 | 91.86 ±1.1 | 89.36 ±1.1 | 90.43 ±0.9 | 97.83 ±1.1 | 73.14 ±3.5 |
| | Greedy | 96.36 | 92.49 | 95.31 | 93.62 | 91.52 | 97.25 | 74.19 |
| | Evo | 94.30 | 90.40 | 93.42 | 92.96 | 94.12 | 99.06* | 76.45 |
| | Core-set | 93.85 | 89.74 | 93.89 | 92.55 | 92.13 | 96.88 | 75.06 |
| **10 samples** | Random | 93.59 ±0.4 | 89.05 ±0.5 | 93.24 ±0.3 | 93.00 ±0.6 | 92.80 ±0.6 | 97.96 ±1.0 | 75.43 ±3.0 |
| | Greedy | 96.37 | 92.59 | 95.38 | 94.91 | 92.94 | 97.06 | 77.80 |
| | Evo | 95.51 | 91.78 | 94.28 | 95.37 | 96.37 | 98.81* | 76.61 |
| | Core-set | 94.28 | 91.03 | 94.47 | 94.94 | 96.77 | 98.19 | 76.78 |
| **25 samples** | Random | 94.68 ±0.2 | 91.52 ±0.2 | 94.16 ±0.2 | 96.24 ±0.3 | 95.29 ±0.3 | 98.09 ±0.6 | 77.08 ±0.9 |
| | Greedy | 96.29 | 92.60 | 95.52 | 96.06 | 93.53 | 97.94 | 78.60* |
| | Evo | 95.51 | 91.88 | 95.07 | 97.30 | **98.52**\* | 98.87* | **80.59**\* |
| | Core-set | 94.85 | 92.88 | 95.13 | 97.33 | 98.37 | **99.12**\* | 79.18* |
| | Full training | **96.58** | **94.92** | **95.79** | **98.41** | 98.48 | 98.75 | 77.97 |

## 5 Results and Discussion

In the following, we first present one of our key findings: how training on just a few carefully selected samples can, surprisingly, outperform training on the entire dataset (Section 5.1). We then explore the nature of the selected samples produced by our core-set selection strategy and demonstrate that these samples represent prototypical "normal" in-distribution examples, while filtering out irrelevant or less informative samples (Section 5.2). Next, we show that this phenomenon is not restricted to a specific model-dataset combination. Instead, the selected samples transfer effectively across different models and datasets, indicating that they are generally good representative samples for the task, rather than being tailored to a particular model or dataset (Section 5.3). Following this, we discuss how the long-tailed distribution of certain datasets may hinder model performance, as not all in-distribution samples contribute equally to training and examine whether some of these samples should be considered outliers (Sections 5.4 and 5.5). Furthermore, we compare the different types of prototypical samples selected by the algorithms, highlighting that there are multiple modes of "normality", and that in some cases, one algorithm may be more suitable than another depending on the dataset (Section 5.6). Finally, we highlight some practical considerations in Section 5.7.

### 5.1 A Few Selected Samples Can Outperform Training With the Whole Dataset

As shown in Table 1, UAD models achieve high performance even when trained with only a few samples.

Surprisingly, even randomly selected subsets can result in a strong model on all considered datasets. Figure 2 depicts a typical observation that detection performance saturates already with very few samples, regardless of the UAD model used. However, not all random subsets perform equally well, which can be seen by the large standard deviations (up to 15.1) for many categories (see Table 2 in Appendix C). Our proposed

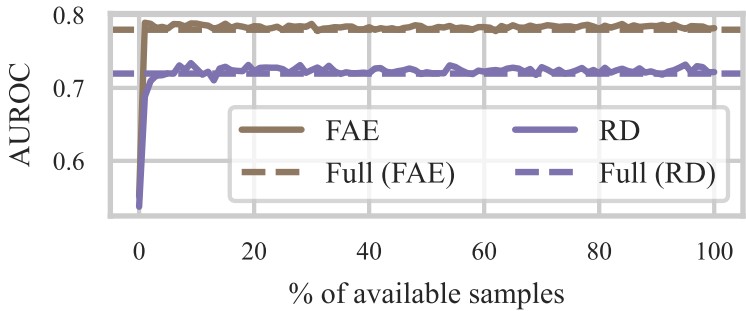

Figure 2: The performance of FAE and RD, after training with an increasing number of samples from RSNA, converges very quickly to the performance when training with the full dataset. The samples are selected according to our core-set selection strategy.

core-set selection strategy substantially outperforms random selection on all datasets and even performs better than full training on BraTS and RSNA. Notably, for the latter, it uses as little as 0.3% of the available training data. In addition to this, the evolutionary algorithm and especially the greedy selection strategy also demonstrate good performance. Moreover, our core-set selection strategy is not far behind and outperforms the other two with 25 samples on CIFAR100 and MNIST despite not using any labels. Overall, our selection strategies outperform full training in 25/67 categories tested in this study (see Tables 2 to 6 in Appendix C). Furthermore, the gap between random and informed selection becomes even more pronounced in the very-low data regime (1–10 samples).

## 5.2    What Characterizes Normal Samples?

Our core-set selection method not only shows that training strong UAD models with only a few samples is possible, but it also provides valuable insights into what constitutes a prototypical in-distribution image. Figure 3 shows the best- and worst-performing samples for each class in CIFAR10 on the left. The "best" images display well-lit prototypical objects that are well-centered, have good contrast, and have mostly uniform backgrounds. In contrast, the "worst" in-distribution images include drawings (bird and horse), toys (frog, truck), historical objects (plane, car), or images with bad contrast (dog). In noisy, less well-curated datasets like RSNA, our method effectively detects and filters low-quality samples. Figure 3 reveals severe deformations, foreign objects such as access tubes or implants, low tissue contrast, or dislocations in the worst-performing samples. The best-performing ones, on the other side, are well-centered and detailed, contain male and female samples, and clearly show the lungs, a prerequisite for detecting pneumonia.

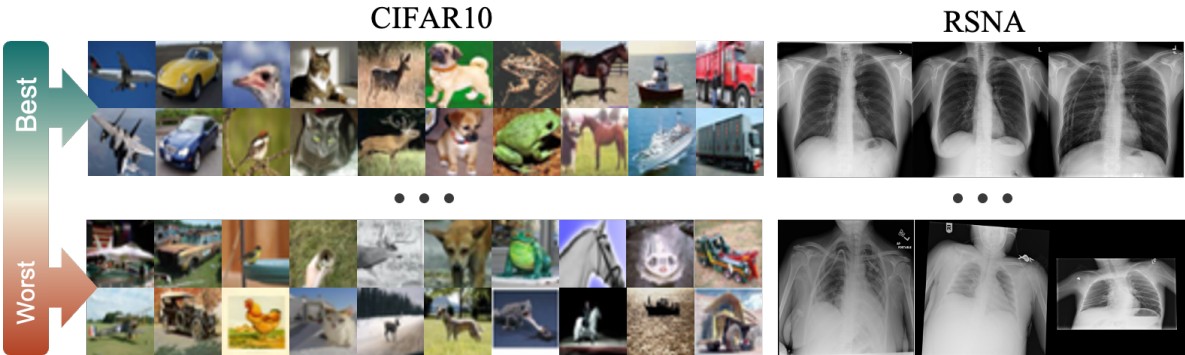

Figure 3: Best- and worst-performing samples in CIFAR10 and RSNA. Identified using our proposed core-set selection strategy.

Motivated by the insights we gained about the characteristics of in-distribution samples, we manually selected a training subset. We chose the RSNA dataset for this experiment because, unlike MVTec-AD, it contains atypical samples in the "normal" data and because the images are large enough to be visually inspected,

in contrast to the other natural image datasets. We selected samples that had similar characteristics as displayed in Figure 3 and covered the distribution of ID samples well. We only started the evaluation of the manually selected samples once their selection was complete. No information other than the characteristics described above was used for the manual selection, and the author selecting the samples is not a trained radiologist or other medical expert. When training with these manually selected samples, we achieved AUROCs of **67**.**88**, **76**.**61**, **79**.**73**, and **81**.**04** for 1, 5, 10, and 25 samples, respectively. The manual selection strategy, therefore, outperformed all automatic selection strategies and even full training, giving the best results on RSNA in this work.

We repeated a similar experiment for the – arbitrarily selected – "8" class on MNIST. This time, however, we did not look at the best- or worst-performing samples from this class but simply selected samples that are visually close to a prototypical 8. We discarded samples that had non-closed lines, where the curves of the 8 were excessively slim, and those with irregular or wavy lines. With these samples, we achieved AUROCs of **76**.**69**, **92**.**42**, **94**.**59**, and **96**.**37** for 1, 5, 10, and 25 samples, respectively, outperforming both random selection and the evolutionary strategy, while also only falling 1.45 points below full training performance.

### 5.3 Prototypical Samples Are Transferrable to Other Models and Datasets

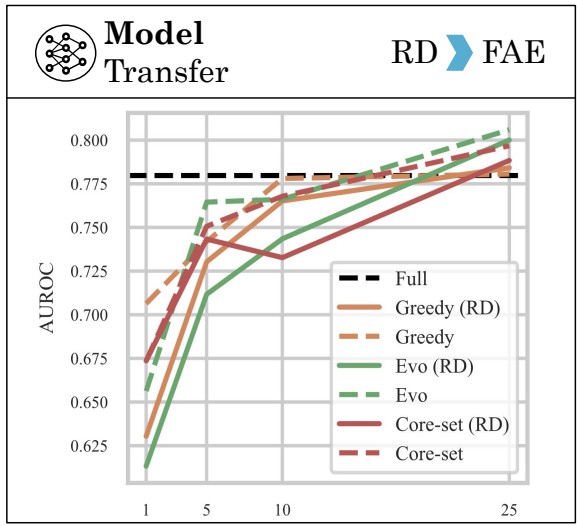
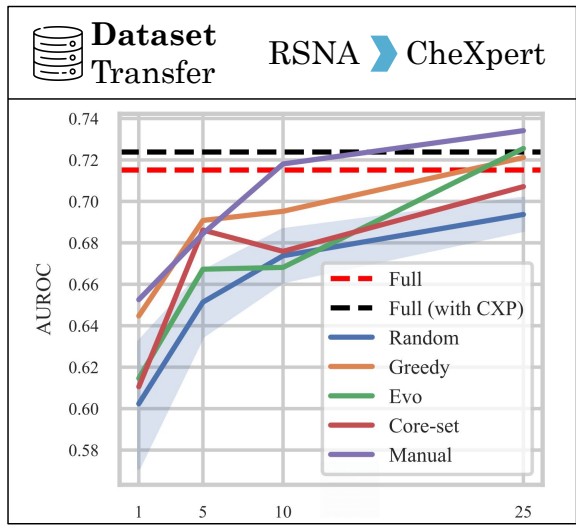

Figure 4: Prototypical samples transfer well to other datasets and models. **Left**: Selected samples with the RD model achieve high performance when used with FAE. Test performance of FAE on RSNA when samples are selected using RD (full lines) or FAE (dashed lines). **Right**: Training with 25 carefully selected samples from RSNA can exceed full training performance with CheXpert (8443 samples) when evaluated on the latter. Test performance on CheXpert after training on CheXpert samples (black, dashed line) or RSNA (other lines).

We also trained a second type of UAD model, Reverse Distillation (RD), on RSNA. This model matches encoder and decoder representations at different levels. The left side of Figure 4 shows how the best-performing samples selected with our proposed core-set selection and the two weakly-supervised baselines on RD perform when applied to the FAE model. Although RD generally performs worse than FAE, its best-performing set of samples works well on FAE, performing almost on par with the ones found with FAE itself and even exceeding full performance. We conclude from this result that there are commonalities between the best samples that are independent of the model. This means that samples found using one model can be transferred to another.

Similarly, high performance for sample combinations on RSNA translates well to the CheXpert dataset. As expected, training with RSNA samples gives a slightly lower performance on CheXpert than training on CheXpert itself (red and black dashed lines in Figure 4, right). This gap, however, can be closed by training

with only 25 high-performing samples from RSNA. Even more impressive, we reached higher performance on CheXpert when training with the 25 manually selected RSNA samples than when training on the full CheXpert dataset itself.

## 5.4 Long-Tail In-Distribution Samples Can Degrade Anomaly Detection Performance

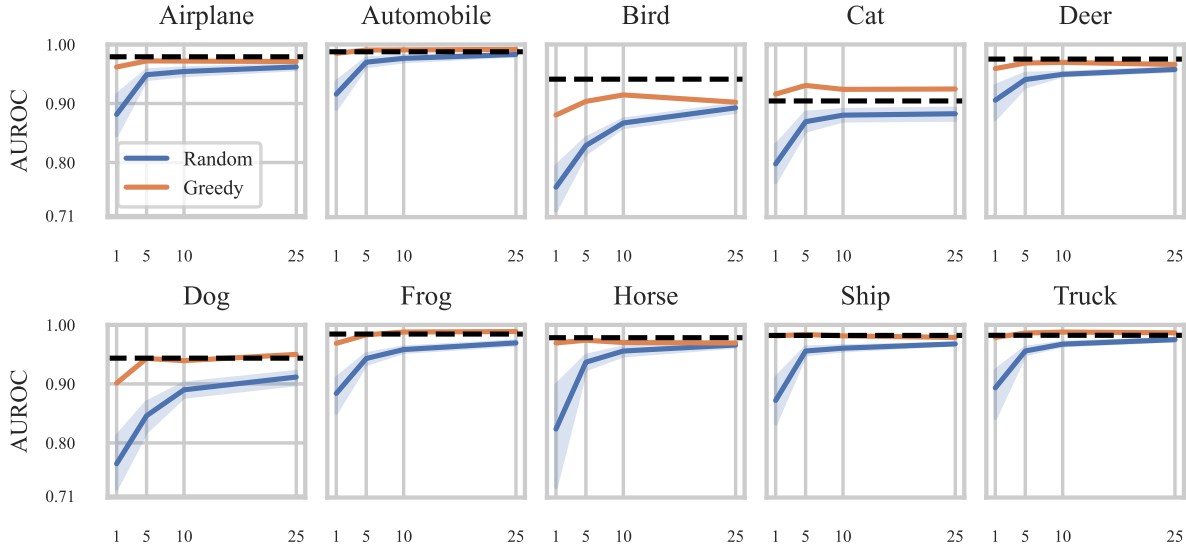

Figure 5: Only 10 representative training samples are needed to surpass the performance of training with the whole dataset on five out of the ten classes in CIFAR10. AUROC for training with a greedy or random subset of maximum size 25. For random, the experiments were repeated ten times with different samples. The dashed black line represents training with all 4000 ID samples.

In Table 1 and figs. 2 and 4, we have seen that very small datasets can exceed the performance of full training. Figure 5 even shows that peak performance for the "cat" class of CIFAR10 is achieved with only five samples. These results suggest that there exist samples in many datasets whose inclusion degrades performance. We hypothesize that the reason for this is due to the nature of the in-distributions, which often have long tails, as shown by Feldman (2020) and Zhu et al. (2014). The long-tail hypothesis states that the majority of the in-distribution samples have only low inter-sample variance, with the exception of a few rare samples that differ a lot from the rest (while still being part of the in-distribution). While these samples can be actively contrasted to other classes and memorized by supervised machine learning models (Feldman & Zhang, 2020), such mechanisms are not available for UAD models, where the long-tail in-distribution samples are treated as any other training sample. This may shift the decision boundary in an unfortunate way (Figure 6, left) and clearly, training with carefully selected samples effectively ignores these data points and can lead to better performance despite using fewer samples (Figure 6, right). An analysis of feature space distances for FAE and RSNA(Figure 7) reveals that the dataset used in our study also follows a long-tail distribution and that the samples at the tails perform worse. Additionally, the worst-performing samples in Figure 3 are clearly atypical and, thus, likely also lay at the tail of the in-distribution.

## 5.5 Should Samples From the Long Tails of the In-Distribution Be Considered Outliers?

Our experiments suggest that there are samples in the training datasets that lie at the tails of the in-distribution and lower the performance of the UAD model. Our subset-selection strategies are effective at filtering out these data points. Of course, ignoring such long-tail samples during training will declare them as outliers, which, at first sight, is false given the labels. We argue, however, that these data points should be considered as such because they warrant special consideration in downstream tasks. For example,

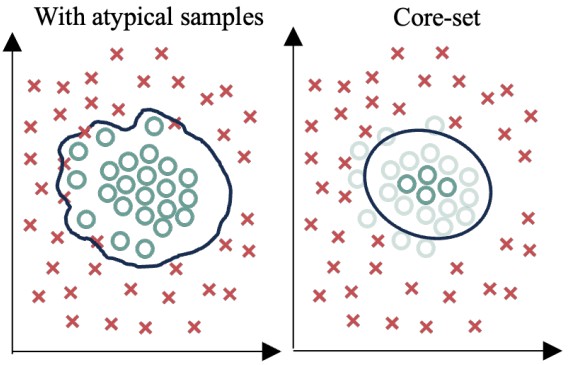

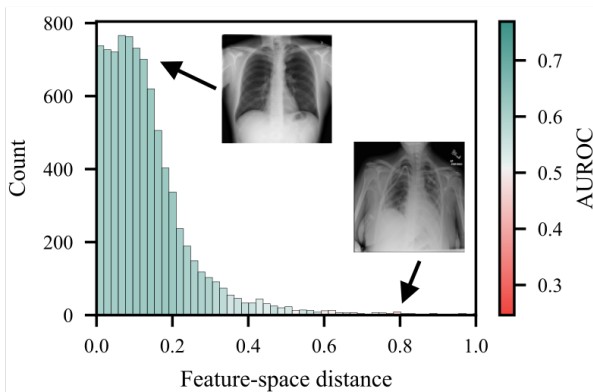

Figure 6: Illustration of our hypothesis of how training with selected samples may prevent in-distribution tail samples from skewing the decision boundary. Left: Some ID samples near OOD data may shift the boundary. Right: Selecting key samples improves the boundary.

Figure 7: RSNA exhibits a long-tail distribution in the FAE feature space, with tail samples showing lower performance. Histogram of distances to the FAE center for all of $X_{train}$, along with two representative images. The bins are colored-coded by the AUROC of each sample, as described in Section 3.1.1.

a subsequent supervised classification algorithm is more likely to misclassify these, and in such a scenario, flagging long tail samples as OOD is desired. Atypical X-ray images, as shown on the bottom right of Figure 3, can pose difficulties (even for manual diagnosis) and should also receive special attention. Further, as we have shown, including these samples during training can lower the classification performance for other samples that are then falsely flagged as ID. The selection methods presented in this study can, therefore, not only be used to identify the most prototypical in-distribution samples but can also be used to automatically filter a dataset from noisy or corrupted images.

## 5.6 The Distributions of Normality and Abnormality Differ Between Datasets

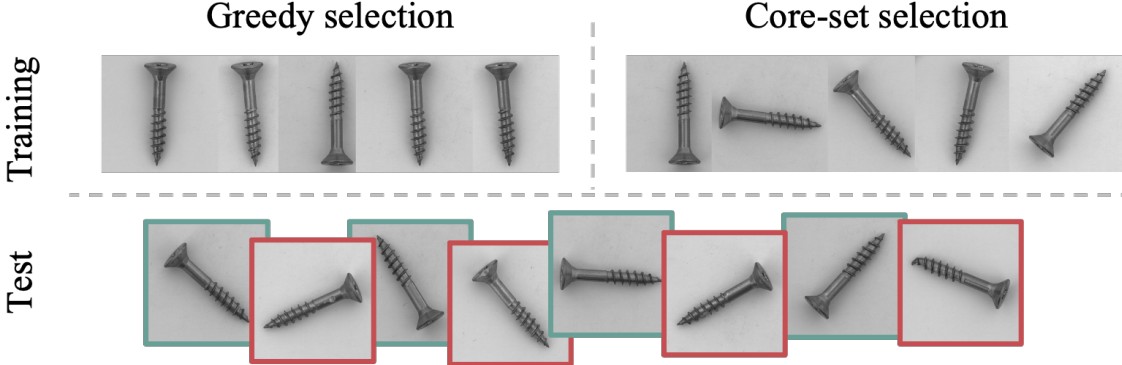

Figure 8: Greedy selection favors visually similar samples, while the core-set selection achieves a better coverage of the normal variations in the training dataset. Best five training samples for the *screw* category in MVTec-AD, found using greedy selection (left) and the unsupervised core-set selection (right). Bottom: normal (green) and defective (red) samples in the test set.

Despite all being used for anomaly detection and showing similar behavior regarding training with few samples, the datasets considered in our study vary greatly with respect to their in- and out-distributions, as well as the relationships between the two. While for some datasets, the variability between ID samples is comparably low; other datasets contain multiple modes of normality. The objects within each class of the MVTec-AD dataset are very similar and often only oriented differently, and the chest X-ray images all show

the same anatomical region. The brain MR images in BraTS are registered to an atlas and, consequently, have even lower anatomical variance. This is in contrast to CIFAR10, CIFAR100, and Fashion-MNIST, where the in-distribution training class can exhibit various shapes, poses, and colors. Similarly, OOD samples can be local and subtle, as in MVTec-AD, BraTS, and RSNA, or global, as in CIFAR10, CIFAR100, MNIST, Fashion-MNIST. A good subset of prototypical samples should cover the different modes of normality but exclude the samples that are atypical and semantically too close to the out-distribution. When looking at examples of the different types of in- and out-distributions described above, we can identify the strengths and weaknesses of the different selection strategies. An example of a dataset with different ID modes and subtle anomalies is the "screw" category of MVTec-AD. Figure 8 shows that greedy selection favors combinations of similarly oriented images and fails to cover the whole space of differently-oriented, normal samples. Our proposed unsupervised core-set selection strategy, on the other hand, (and also the evolutionary algorithm) better covers the different orientations that should all be considered normal (see also Table 6 in Appendix C).

### 5.7 Limitations and Practical Considerations

In this work, we introduced three different subset selection methods: our proposed unsupervised core-set selection approach, as well as two weakly supervised baselines—a greedy algorithm and an evolutionary algorithm—that leverage labels from the validation set. Unlike the weakly supervised baselines, our unsupervised core-set selection method has the potential for practical application, as it enables the derivation of significantly smaller training datasets while maintaining strong performance. However, its behavior is not perfectly stable, as test performance may sometimes fall below, be on par with, or exceed that of full-dataset training, indicating that further refinement is necessary (see Section 6). With further improvements, this approach could be a viable method for selecting efficient training subsets in real-world UAD scenarios and offer valuable insights into what constitutes "normal" in-distribution samples.

In contrast, the weakly supervised methods were not designed as practical approaches but rather as benchmark baselines for comparison. A key practical limitation of these baselines is their computational cost, as they require training a model on each individual sample in the dataset. This approach becomes infeasible for large datasets and is not recommended in practice. More importantly, in a true unsupervised setting, where labels are unavailable, such weakly supervised methods cannot be used to select an optimal training subset, further reinforcing their role as comparative baselines rather than deployable solutions.

## 6 Future Work

Our results highlight the potential of unsupervised core-set selection for training anomaly detection models with significantly reduced datasets while maintaining strong performance. While our findings are an important first step towards understanding this phenomenon and potentially provide a new way of constructing datasets for UAD tasks, we emphasise that more research is required for a well-founded understanding. Future work should focus on refining our core-set selection algorithm to enhance its ability to extract the most representative and informative training samples, while also exploring alternative core-set selection methods that may provide even better results. Since our approach sometimes performs slightly worse, the same, or even better than training on the full dataset, optimizing the selection process could lead to practical applications where smaller, high-quality training sets provide an efficient alternative to large datasets. Given the advantages of working with smaller datasets—such as easier (and cheaper) acquisition, applicability to a wider range of tasks, and reduced training costs—further research could explore strategies to enhance the robustness and consistency of this method. Additionally, it is crucial to extend this investigation to other datasets and models, to fully explore the generalizability of the discovered phenomenon and identify whether the observed effects hold in different contexts.

A crucial consideration in future research is the potential loss of important information when removing large portions of the dataset. For example in the medical domain, this raises ethical concerns, particularly regarding fairness and representation. If minority group samples are unintentionally excluded from the training dataset due to an unbalanced split, the model may fail to generalize properly at inference time when encountering these underrepresented groups. Developing selection methods that not only optimize

performance but also preserve all critical information in the training samples is a key direction for future work.

Beyond algorithmic improvements, future studies should continue investigating the characteristics of selected core-sets, particularly in relation to long-tailed distributions and the existence of multiple modes of normality. Understanding which samples contribute most to training performance and which hinder it can provide deeper insights into dataset quality and UAD model behavior.

## 7 Conclusion

In many areas of deep learning, it is widely believed that more training data leads to improved model performance. Our work challenges this assumption within the context of Unsupervised Anomaly Detection (UAD) and strongly emphasizes that data quality should be prioritized over quantity. Specifically, we demonstrate that UAD models can achieve state-of-the-art performance with surprisingly few training samples ($\leq 25$), underscoring that less can often be more when it comes to training data. We introduce a novel, unsupervised core-set selection strategy that efficiently and automatically extracts high-performing, prototypical training samples. Our approach is fast, easy to implement, and shows that these prototypical in-distribution samples are transferable across different models, datasets, modalities, and tasks. Furthermore, we highlight that the key characteristics of these samples enable effective manual selection, offering insights into the data points that drive optimal model performance. Ultimately, our results show that UAD models can be more efficient, easier to train, and significantly more practical in real-world applications, even though their performance may still fall short of supervised methods.

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

# A    Appendix: Datasets

### A.0.1   CIFAR10, CIFAR100, MNIST, and FashionMNIST

We follow the widely used training setup from Reiss et al. (2021) for these datasets. CIFAR10 contains 6000 images per class. For each class $c \in C$, we create a training dataset $X_{\text{train, c}}$ using 4000 samples from said class. The remaining samples are split equally into a validation and a test set and combined with the same amount of samples from every other class as outliers. Training, validation, and test sets for CIFAR100, MNIST, and Fashion-MNIST are created analogously. For CIFAR100, we used the 20 superclasses instead of the 100 detailed classes.

### A.0.2   MVTec-AD

MVTec-AD is a dataset for defect detection in industrial production. Here, we used the original splits as outlined by Bergmann et al. (2019).

### A.0.3   BraTS

The multimodal brain tumor image segmentation benchmark (BraTS) contains 369 MRI from patients with glioma. We extract 5 slices around the center line and use 101 slices without glioma for training. The remaining 80 normal slices are split 50:50 into a validation and a test set and are complemented with 40 pathological slices each. Following Meissen et al. (2023), we use the T2-weighted sequences, perform histogram equalization on the slices, and resize them to $128 \times 128$.

### A.0.4   RSNA and CheXpert

The RSNA Pneumonia Detection dataset (Stein et al., 2018) is a subset of 30 000 frontal view chest radiographs from the National Institutes of Health (NIH) CXR8 dataset that was manually labeled by 18 radiologists for one of the following labels: "Normal", "Lung Opacity", or "No Lung Opacity / Not Normal". The CheXpert database contains 224 316 chest-radiographs of 65 240 patients, acquired at Stanford Hospital with 13 structured diagnostic labels. To make the CheXpert compatible with RSNA, we only considered frontal-view images without support devices and further excluded those where any of the labels were marked as uncertain. In addition to the image data and labels, demographic information about the patients' gender and age was available for both datasets.

For RSNA, we used the "Normal" label as in-distribution images and combined the "pneumonia" and "No Opacity/Not Normal" (has lung opacities, but not suspicious for pneumonia) as OOD. Similarly, for CheXpert, the "No Finding" label was used as in-distribution, and "pneumonia" and "lung opacity" as OOD. For both datasets, we created a validation- and a test set that are balanced w.r.t. gender (male and female), age (young and old), the presence of anomalies, and contain 800 samples each. The remaining in-distribution samples were used for training. As part of preprocessing, all Chest X-ray images were center cropped and resized to $128 \times 128$ pixels. Note that we treated both datasets individually and did not combine them for training or evaluation.

# B    Appendix: Models

### B.0.1   PANDA

PANDA (Reiss et al., 2021) is a model built upon DeepSVDD by Ruff et al. (2018). Like the latter, it relies on training a one-class classifier by using the compactness loss. Given a feature-extractor $\psi$ and a center vector $c$, the compactness loss is defined as:

$$\mathcal{L}_{\text{compact}} = \sum_{x \in D} ||\psi(x) - c||^2 . \tag{5}$$

The center vector $c$ is computed as the mean feature vector of the training dataset $D$ on the untrained model $\psi_0$:

$$c = \frac{1}{|D|} \sum_{x \in D} \psi_0(x).$$ (6)

Instead of training a specialized architecture from scratch, PANDA benefits from useful features of pre-trained models. Specifically, it extracts features from the penultimate layer of a ResNet152 (He et al., 2016) and categorizes samples into ID / OOD by the L1-distance to the $k$ nearest neighbors (kNN), with $k = 2$. PANDA only fine-tunes `layer3` and `layer4` and uses early stopping to determine the optimal distance between ID and OOD samples. Training has no effect when using only one sample as the feature representation of the only sample is always identical to $c$. We used the original configuration from their paper for our experiments.

### B.0.2 PatchCore

PatchCore (Roth et al., 2022) follows a similar concept as PANDA. However, instead of performing a kNN search on pooled, global features, PatchCore achieves localized anomaly detection by using a memory bank of locally-aware patch features $\mathcal{M}$ instead. To make the kNN search computationally feasible, PatchCore performs core-set selection on the memory bank to retain only a subset of representative features $\mathcal{M}_{\text{sub}}$:

$$\mathcal{M}_{\text{sub}}^* = \arg\min_{\mathcal{M}_{\text{sub}} \subset \mathcal{M}} \max_{m \in \mathcal{M}} \min_{n \in \mathcal{M}_{\text{sub}}} ||m - n||_2.$$ (7)

Compared to PANDA, PatchCore does not require fine-tuning of the feature extractor. We used the same hyperparameters for PatchCore as in the original publication.

### B.0.3 FAE

The Structural Feature-Autoencoder (FAE) (Meissen et al., 2023) extracts spatial feature maps from a pre-trained and frozen feature extractor $\psi$ (a ResNet18 He et al. (2016) in practice). The feature maps a resized and concatenated and fed into a convolutional autoencoder $f_\theta$ that is trained via the structural similarity (SSIM) loss for reconstruction:

$$\mathcal{L} = \text{SSIM}(\psi(x), f_\theta(\psi(x))).$$ (8)

Anomalies are detected using the residual between the feature maps and their reconstruction like in popular image-reconstruction models. It was found to perform best for UAD in Chest X-ray images in a study by Lagogiannis et al. (2023). We use a smaller model in our experiments since it still gave us the same performance on the RSNA full dataset as the larger model and is more resource-efficient. Specifically, we set the `fae_hidden_dims` parameter to `[50, 100]`. The other parameters were kept the same as in the original publication.

### B.0.4 Reverse Distillation

The Reverse Distillation (RD) model by Deng & Li (2022) utilizes a frozen encoder model and a decoder that mirrors the former, similar to an Autoencoder. Instead of reconstructing the image, however, RD minimizes the cosine distance between feature maps in the encoder and decoder. The same measure is also used during inference to detect anomalies. RD was the second-best performing UAD model for Chest X-ray images in Lagogiannis et al. (2023). We used the same hyperparameters for RD as in Lagogiannis et al. (2023).

## C   Appendix: Detailed results

We show the detailed per-class results for CIFAR10, CIFAR100, MNIST, Fashion-MNIST, and MVTec-AD in Tables 2 to 6, respectively.

Table 2: Detailed training results for PANDA on CIFAR-10. AUROC scores are reported for full training and training on subsets of 1, 5, 10, and 25 samples, selected using either our proposed unsupervised core-set selection method, weakly supervised greedy search and evolutionary algorithms, or random sampling. The numbers for random selection include ± values indicating the standard deviation across ten different randomly drawn subsets. Best overall performances are marked in **bold**, while underlined numbers indicate the best performance for each sample size.

| | Method | Airplane | Automobile | Bird | Cat | Deer | Dog | Frog | Horse | Ship | Truck | Average |
|---|---|---|---|---|---|---|---|---|---|---|---|---|
| 1 sample | Random | 88.14 ±6.3 | 91.59 ±4.1 | 75.81 ±6.2 | 79.75 ±5.5 | 90.55 ±4.7 | 76.47 ±7.6 | 88.38 ±5.1 | 82.34 ±15.1 | 87.19 ±6.6 | 89.32 ±7.6 | 84.96 ±1.2 |
| | Greedy | 96.18 | 98.50 | 88.04 | 91.59 | 95.92 | 90.13 | 96.87 | 96.94 | 98.14 | 97.88 | 95.02 |
| | Evo | 86.15 | 97.08 | 88.04 | 78.61 | 93.27 | 79.83 | 67.29 | 94.21 | 96.64 | 95.73 | 87.68 |
| | Core-set | 94.54 | 96.86 | 79.14 | 81.56 | 92.75 | 84.63 | 92.84 | 92.28 | 92.09 | | 90.08 |
| 5 samples | Random | 94.88 ±1.3 | 97.00 ±1.2 | 82.88 ±2.4 | 86.92 ±2.7 | 94.06 ±2.0 | 84.61 ±4.5 | 94.30 ±1.5 | 93.68 ±2.0 | 95.57 ±1.1 | 95.58 ±1.0 | 91.95 ±0.7 |
| | Greedy | 97.22 | 99.05 | 90.36 | **93.06** | 96.86 | 94.32 | 98.31 | 97.39 | 98.37 | 98.61 | 96.36 |
| | Evo | 95.85 | 96.60 | 89.19 | 87.44 | 96.12 | 92.23 | 93.89 | 97.11 | 96.72 | 97.88 | 94.30 |
| | Core-set | 94.14 | 97.23 | 88.70 | 87.88 | 95.91 | 90.36 | 95.24 | 96.39 | 96.22 | 96.46 | 93.85 |
| 10 samples | Random | 95.42 ±1.2 | 97.65 ±0.8 | 86.70 ±1.2 | 88.02 ±1.6 | 94.94 ±0.5 | 89.00 ±2.0 | 95.82 ±0.8 | 95.57 ±1.3 | 96.05 ±0.7 | 96.74 ±0.5 | 93.59 ±0.4 |
| | Greedy | 97.19 | 99.08 | 91.45 | 92.39 | 96.97 | 93.94 | 98.79 | 97.01 | 98.14 | **98.79** | 96.37 |
| | Evo | 96.47 | 97.98 | 90.64 | 89.32 | 96.62 | 93.61 | 97.72 | 96.93 | 97.62 | 98.23 | 95.51 |
| | Core-set | 94.90 | 97.79 | 88.56 | 88.09 | 95.91 | 91.50 | 96.31 | 96.46 | 96.45 | 96.81 | 94.28 |
| 25 samples | Random | 96.19 ±0.7 | 98.30 ±0.3 | 89.26 ±1.2 | 88.26 ±1.7 | 95.77 ±0.5 | 91.16 ±1.9 | 96.95 ±0.6 | 96.59 ±0.5 | 96.77 ±0.2 | 97.50 ±0.3 | 94.68 ±0.2 |
| | Greedy | 97.11 | **99.17** | 90.22 | 92.46 | 96.61 | **95.00** | **98.87** | 96.93 | 97.87 | 98.65 | 96.29 |
| | Evo | 96.56 | 98.18 | 91.38 | 89.16 | 96.68 | 92.11 | 97.87 | 97.42 | 97.64 | 98.08 | 95.51 |
| | Core-set | 96.88 | 98.67 | 89.06 | 86.26 | 95.98 | 92.98 | 97.40 | 96.99 | 96.68 | 97.56 | 94.85 |
| | Full training | **97.92** | 98.76 | **94.12** | 90.43 | **97.53** | 94.36 | 98.44 | **97.83** | 98.21 | 98.23 | **96.58** |

Table 3: Detailed training results for PANDA on CIFAR-100. AUROC scores are reported for full training and training on subsets of 1, 5, 10, and 25 samples, selected using either our proposed unsupervised core-set selection method, weakly supervised greedy search and evolutionary algorithms, or random sampling. The numbers for random selection include ± values indicating the standard deviation across ten different randomly drawn subsets. Best overall performances are marked in **bold**, while underlined numbers indicate the best performance for each sample size.

| | Method | Aquatic mammals | Fish | Flowers | Food containers | Fruit and vegetables | Household electrical devices | Household furniture | Insects | Large carnivores | Large man-made outdoor things |
|---|---|---|---|---|---|---|---|---|---|---|---|
| 1 sample | Random | 71.70 ±14.7 | 75.23 ±8.1 | 90.83 ±3.9 | 82.25 ±3.7 | 71.29 ±12.5 | 71.88 ±7.1 | 84.70 ±6.1 | 68.09 ±12.3 | 77.95 ±11.7 | 86.72 ±6.0 |
| | Greedy | 91.93 | 89.09 | 96.78 | 89.39 | 90.97 | 85.73 | 96.17 | 84.78 | 92.46 | 93.98 |
| | Evo | 89.68 | 78.96 | 80.31 | 76.69 | 84.76 | 80.61 | 92.09 | 63.07 | 85.80 | 90.19 |
| | Core-set | 64.34 | 75.71 | 90.52 | 80.05 | 85.14 | 73.08 | 90.44 | 61.80 | 68.79 | 89.97 |
| 5 samples | Random | 86.25 ±3.8 | 81.82 ±6.1 | 95.82 ±1.4 | 89.31 ±2.4 | 85.26 ±5.9 | 78.90 ±6.2 | 92.54 ±2.0 | 79.26 ±9.0 | 87.47 ±1.7 | 92.15 ±1.5 |
| | Greedy | **94.13** | 92.32 | 98.13 | 95.59 | 93.00 | 85.05 | 96.78 | 91.05 | 93.73 | 95.55 |
| | Evo | 93.04 | 89.85 | 97.08 | 85.89 | 92.47 | 89.12 | 95.64 | 86.96 | 90.11 | 93.78 |
| | Core-set | 86.53 | 88.90 | 96.48 | 91.79 | 91.29 | 86.67 | 95.75 | 87.01 | 91.77 | 92.95 |
| 10 samples | Random | 88.50 ±1.9 | 85.56 ±4.8 | 96.97 ±0.7 | 91.31 ±1.7 | 90.59 ±3.1 | 80.97 ±4.3 | 94.77 ±0.8 | 81.96 ±5.0 | 90.22 ±1.4 | 92.49 ±1.2 |
| | Greedy | 93.84 | 91.92 | **98.60** | **95.97** | 92.88 | 82.56 | 96.81 | 89.68 | 93.33 | 95.57 |
| | Evo | 93.19 | 89.98 | 98.16 | 91.44 | 93.24 | 88.27 | 96.86 | 88.92 | 90.95 | 93.79 |
| | Core-set | 91.31 | 90.85 | 97.41 | 92.08 | 92.28 | 86.71 | 95.99 | 85.73 | 92.60 | 94.00 |
| 25 samples | Random | 90.04 ±1.3 | 90.78 ±1.2 | 97.87 ±0.5 | 93.19 ±1.5 | 93.63 ±0.6 | 84.21 ±4.1 | 96.28 ±0.5 | 86.34 ±1.7 | 91.82 ±0.8 | 93.50 ±0.6 |
| | Greedy | 92.36 | 91.27 | 98.50 | 95.79 | 92.53 | 80.51 | 97.26 | 90.89 | 93.21 | 95.45 |
| | Evo | 92.88 | 92.50 | 98.39 | 92.87 | 95.00 | 90.37 | 96.84 | 90.74 | 91.84 | 94.29 |
| | Core-set | 91.72 | 91.87 | 98.23 | 94.68 | 94.94 | 90.80 | 96.21 | 90.31 | 93.09 | 93.92 |
| | Full training | 93.68 | **94.81** | 98.46 | 95.86 | **96.59** | **94.59** | **97.26** | **93.00** | **95.34** | 94.98 |

| | Method | Large natural outdoor scenes | Large omnivores and herbivores | Medium-sized mammals | Non-insect invertebrates | People | Reptiles | Small mammals | Trees | Vehicles 1 | Vehicles 2 |
|---|---|---|---|---|---|---|---|---|---|---|---|
| 1 sample | Random | 86.87 ±6.3 | 66.79 ±14.8 | 73.91 ±7.6 | 60.08 ±8.3 | 88.73 ±6.4 | 66.99 ±7.5 | 72.46 ±9.3 | 92.30 ±3.6 | 78.59 ±8.2 | 73.74 ±6.6 |
| | Greedy | 94.13 | 86.01 | 84.57 | 79.51 | 96.20 | 79.39 | 88.86 | 95.92 | 93.20 | 89.81 |
| | Evo | 88.88 | 78.12 | 73.68 | 77.34 | 83.52 | 52.97 | 86.70 | 95.58 | 68.73 | 72.15 |
| | Core-set | 93.57 | 74.96 | 79.55 | 56.37 | 91.10 | 72.36 | 87.23 | 94.28 | 82.56 | 80.49 |
| 5 samples | Random | 92.30 ±2.0 | 82.54 ±3.3 | 84.66 ±2.1 | 71.37 ±5.1 | 95.66 ±1.2 | 79.22 ±3.2 | 85.18 ±4.6 | 95.67 ±0.9 | 88.96 ±3.2 | 83.75 ±3.8 |
| | Greedy | 94.98 | 90.29 | 89.08 | 82.67 | 97.90 | 84.47 | 92.49 | 96.89 | 95.61 | 90.10 |
| | Evo | 95.11 | 87.96 | 83.67 | 81.76 | 97.33 | 81.79 | 90.32 | 96.83 | 92.12 | 87.22 |
| | Core-set | 93.92 | 86.92 | 87.56 | 75.77 | 95.14 | 85.19 | 86.94 | 95.48 | 92.46 | 86.36 |
| 10 samples | Random | 93.76 ±1.2 | 85.27 ±2.7 | 87.79 ±1.3 | 78.13 ±2.9 | 96.39 ±0.8 | 83.14 ±3.0 | 87.63 ±2.7 | 96.44 ±0.5 | 92.30 ±1.8 | 86.80 ±1.5 |
| | Greedy | 95.49 | 89.41 | 90.43 | 85.01 | 98.18 | 87.55 | 91.98 | 97.67 | 95.92 | 88.95 |
| | Evo | 94.95 | 89.90 | 88.72 | 83.32 | 97.69 | 85.42 | 90.41 | 97.48 | 94.83 | 88.14 |
| | Core-set | 95.81 | 86.39 | 87.80 | 79.06 | 96.93 | 86.52 | 89.47 | 96.81 | 93.88 | 89.05 |
| 25 samples | Random | 94.93 ±0.7 | 89.69 ±1.5 | 89.68 ±0.6 | 83.07 ±1.6 | 97.47 ±0.5 | 86.95 ±1.1 | 89.92 ±0.8 | 97.06 ±0.3 | 94.09 ±0.4 | 89.90 ±1.0 |
| | Greedy | **95.84** | 90.23 | 90.78 | 86.55 | 98.13 | 89.72 | 91.81 | 97.50 | 95.87 | 87.78 |
| | Evo | 95.44 | 91.58 | 91.27 | 85.68 | 97.97 | 88.61 | 91.33 | 97.17 | 95.64 | 90.17 |
| | Core-set | 95.81 | 91.72 | 90.12 | 83.86 | 97.56 | 88.04 | 90.91 | **97.84** | 94.60 | 91.28 |
| | Full training | 95.37 | **94.40** | **93.54** | 90.76 | **98.55** | 91.71 | 92.70 | 96.84 | **96.06** | **93.88** |

Table 4: Detailed training results for PANDA on MNIST. AUROC scores are reported for full training and training on subsets of 1, 5, 10, and 25 samples, selected using either our proposed unsupervised core-set selection method, weakly supervised greedy search and evolutionary algorithms, or random sampling. The numbers for random selection include ± values indicating the standard deviation across ten different randomly drawn subsets. Best overall performances are marked in **bold**, while underlined numbers indicate the best performance for each sample size.

| | Method | 0 | 1 | 2 | 3 | 4 | 5 | 6 | 7 | 8 | 9 | Average |
|---|---|---|---|---|---|---|---|---|---|---|---|---|
| 1 sample | Random | 91.11 ±2.4 | 90.34 ±6.0 | 65.18 ±10.6 | 77.02 ±4.4 | 69.79 ±11.2 | 69.69 ±9.7 | 69.43 ±6.9 | 70.37 ±7.6 | 76.91 ±3.6 | 81.26 ±5.4 | 76.11 ±2.2 |
| | Greedy | 98.14 | 97.09 | 82.74 | 89.92 | 92.94 | 88.69 | 85.01 | 87.87 | 89.59 | 89.53 | 90.15 |
| | Evo | 82.31 | 83.30 | 69.87 | 75.18 | 78.13 | 82.36 | 78.57 | 59.38 | 62.56 | 76.56 | 74.82 |
| | Core-set | 94.71 | 92.27 | 71.53 | 86.66 | 80.33 | 78.28 | 75.39 | 76.65 | 82.48 | 84.61 | 82.29 |
| 5 samples | Random | 98.52 ±0.9 | 97.79 ±2.2 | 77.40 ±7.3 | 89.79 ±1.9 | 90.25 ±1.9 | 85.58 ±4.9 | 88.90 ±4.3 | 86.97 ±2.6 | 87.92 ±3.5 | 90.42 ±2.1 | 89.36 ±1.1 |
| | Greedy | 99.21 | 98.29 | 85.59 | 93.13 | 95.65 | 90.90 | 93.00 | 92.52 | 94.31 | 93.56 | 93.62 |
| | Evo | 98.76 | 96.96 | 85.06 | 94.82 | 95.78 | 91.44 | 95.57 | 89.28 | 87.35 | 94.59 | 92.96 |
| | Core-set | 99.29 | 99.58 | 86.83 | 92.32 | 89.07 | 91.30 | 94.72 | 91.68 | 91.88 | 88.82 | 92.55 |
| 10 samples | Random | 98.86 ±0.6 | 99.18 ±0.4 | 83.04 ±4.0 | 92.39 ±2.0 | 93.82 ±2.3 | 90.84 ±1.3 | 92.81 ±2.8 | 93.28 ±1.2 | 92.34 ±1.8 | 93.40 ±1.1 | 93.00 ±0.6 |
| | Greedy | 99.43 | 98.83 | 89.00 | 93.99 | 96.89 | 91.39 | 95.62 | 93.97 | 96.28 | 93.75 | 94.91 |
| | Evo | 99.40 | 99.30 | 88.08 | 94.42 | 97.97 | 92.40 | 97.59 | 97.28 | 92.17 | 95.09 | 95.37 |
| | Core-set | 99.38 | 99.66 | 90.60 | 94.23 | 96.27 | 91.46 | 95.65 | 93.56 | 93.45 | 95.16 | 94.94 |
| 25 samples | Random | 99.38 ±0.2 | 99.70 ±0.1 | 90.26 ±2.1 | 95.36 ±1.5 | 96.66 ±1.0 | 93.37 ±1.0 | 97.55 ±1.3 | 97.49 ±0.5 | 95.83 ±0.6 | 96.79 ±0.5 | 96.24 ±0.3 |
| | Greedy | 99.67 | 99.30 | 89.85 | 96.63 | 97.67 | 90.84 | 98.32 | 94.88 | **97.86** | 95.60 | 96.06 |
| | Evo | 99.44 | 99.72 | 91.65 | 97.09 | **99.01** | 95.10 | **99.23** | 98.70 | 95.43 | 97.60 | 97.30 |
| | Core-set | 99.76 | 99.71 | 92.42 | 96.22 | 97.04 | 95.96 | 98.89 | 98.66 | 97.37 | 97.23 | 97.33 |
| | Full training | **99.84** | **99.93** | **96.58** | **98.02** | 98.63 | **96.45** | 99.13 | **99.14** | 97.82 | **98.53** | **98.41** |

Table 5: Detailed training results for PANDA on Fashion MNIST. AUROC scores are reported for full training and training on subsets of 1, 5, 10, and 25 samples, selected using either our proposed unsupervised core-set selection method, weakly supervised greedy search and evolutionary algorithms, or random sampling. The numbers for random selection include ± values indicating the standard deviation across ten different randomly drawn subsets. Best overall performances are marked in **bold**, while underlined numbers indicate the best performance for each sample size.

| | Method | 0 | 1 | 2 | 3 | 4 | 5 | 6 | 7 | 8 | 9 | Average |
|---|---|---|---|---|---|---|---|---|---|---|---|---|
| 1 sample | Random | 77.24 ±11.0 | 96.45 ±0.8 | 82.47 ±17.5 | 73.80 ±7.5 | 75.09 ±12.5 | 91.63 ±4.1 | 69.29 ±12.4 | 94.71 ±6.2 | 79.04 ±8.4 | 95.37 ±4.6 | 83.51 ±3.3 |
| | Greedy | 93.85 | 98.12 | 93.03 | 88.85 | 90.46 | 97.14 | 84.55 | 98.84 | 95.99 | 99.16 | 94.00 |
| | Evo | 89.28 | 83.19 | 88.71 | 83.60 | 79.98 | 94.35 | 71.09 | 96.53 | 84.16 | 97.30 | 86.82 |
| | Core-set | 90.94 | 97.12 | 91.63 | 83.53 | 88.62 | 95.75 | 76.01 | 98.67 | 94.42 | 98.00 | 91.47 |
| 5 samples | Random | 90.58 ±3.7 | 97.90 ±0.5 | 92.13 ±1.8 | 85.24 ±5.4 | 88.77 ±1.3 | 95.38 ±1.9 | 80.89 ±2.4 | 98.34 ±0.8 | 91.01 ±6.7 | 98.41 ±1.2 | 91.86 ±1.1 |
| | Greedy | 94.64 | 98.98 | 94.27 | 92.38 | 92.95 | 97.98 | 86.42 | 99.18 | 96.84 | 99.42 | 95.31 |
| | Evo | 92.74 | 98.85 | 93.94 | 89.42 | 88.10 | 94.19 | 83.62 | 98.59 | 96.30 | 98.43 | 93.42 |
| | Core-set | 92.05 | 97.95 | 92.91 | 91.24 | 89.85 | 97.32 | 84.05 | 98.61 | 96.10 | 98.87 | 93.89 |
| 10 samples | Random | 91.78 ±2.1 | 98.31 ±0.5 | 93.22 ±0.9 | 89.62 ±2.4 | 89.87 ±0.7 | 96.24 ±1.3 | 81.83 ±1.9 | 98.57 ±0.7 | 94.31 ±1.6 | 98.62 ±0.7 | 93.24 ±0.3 |
| | Greedy | 94.79 | 98.98 | 94.53 | 93.38 | 92.00 | **98.55** | 86.74 | 99.27 | 96.02 | 99.50 | 95.38 |
| | Evo | 91.41 | 99.31 | 94.57 | 94.72 | 90.11 | 95.62 | 82.86 | 98.93 | 96.90 | 98.37 | 94.28 |
| | Core-set | 93.77 | 98.85 | 94.14 | 92.82 | 91.40 | 96.13 | 84.14 | 98.72 | 95.98 | 98.70 | 94.47 |
| 25 samples | Random | 93.28 ±1.2 | 98.62 ±0.3 | 93.74 ±0.7 | 93.37 ±1.6 | 91.48 ±0.9 | 95.86 ±1.0 | 82.45 ±1.4 | 98.89 ±0.2 | 95.31 ±1.0 | 98.62 ±0.5 | 94.16 ±0.2 |
| | Greedy | 95.06 | 99.03 | **94.64** | 94.49 | 92.97 | 98.54 | **86.76** | 99.26 | 94.86 | **99.60** | 95.52 |
| | Evo | 94.10 | 99.24 | 94.12 | 95.15 | 92.97 | 95.70 | 84.11 | 98.94 | 97.32 | 99.00 | 95.07 |
| | Core-set | 94.40 | 99.18 | 94.16 | 94.70 | 91.64 | 97.07 | 84.80 | 99.13 | 97.37 | 98.87 | 95.13 |
| | Full training | **95.09** | **99.52** | 94.44 | **96.28** | **93.66** | 96.58 | 85.33 | **99.28** | **98.88** | 98.86 | **95.79** |

Table 6: Detailed training results for PatchCore on MVTec-AD. AUROC scores are reported for full training and training on subsets of 1, 5, 10, and 25 samples, selected using either our proposed unsupervised core-set selection method, weakly supervised greedy search and evolutionary algorithms, or random sampling. The numbers for random selection include ± values indicating the standard deviation across ten different randomly drawn subsets. Best overall performances are marked in **bold**, while underlined numbers indicate the best performance for each sample size.

| | Method | Bottle | Cable | Capsule | Carpet | Grid | Hazelnut | Leather | Metal nut |
|---|---|---|---|---|---|---|---|---|---|
| **1 sample** | Random | 99.71 ±0.1 | 83.74 ±4.7 | 66.80 ±5.1 | 97.72 ±0.5 | 60.30 ±6.9 | 90.67 ±2.6 | 99.99 ±0.0 | 71.99 ±4.0 |
| | Greedy | 99.76 | 88.19 | 72.80 | 98.60 | 71.19 | 93.93 | **100.00** | 72.19 |
| | Evo | 99.52 | 88.92 | 63.14 | 98.23 | 66.88 | 93.04 | **100.00** | 70.97 |
| | Core-set | 99.52 | 89.81 | 61.55 | **99.08** | 65.99 | 88.68 | **100.00** | 71.07 |
| **5 samples** | Random | 99.84 ±0.2 | 91.40 ±3.6 | 84.11 ±7.2 | 97.98 ±0.3 | 72.64 ±7.1 | 96.29 ±2.2 | **100.00** ±0.0 | 94.93 ±3.5 |
| | Greedy | **100.00** | 96.27 | 84.96 | 98.64 | 73.52 | 98.29 | **100.00** | 97.75 |
| | Evo | 99.52 | 93.22 | 90.91 | 98.19 | 86.04 | 99.14 | **100.00** | 98.34 |
| | Core-set | 99.68 | 92.82 | 87.36 | 98.72 | 69.40 | 98.64 | **100.00** | 96.82 |
| **10 samples** | Random | 99.90 ±0.2 | 93.49 ±1.8 | 90.29 ±2.3 | 98.04 ±0.3 | 80.91 ±5.7 | 98.99 ±0.7 | **100.00** ±0.0 | 97.72 ±1.6 |
| | Greedy | **100.00** | 97.58 | 93.06 | 98.15 | 78.86 | 99.64 | **100.00** | 98.92 |
| | Evo | **100.00** | 96.42 | 91.26 | 98.39 | 94.40 | **100.00** | **100.00** | 99.07 |
| | Core-set | 99.60 | 92.75 | 90.75 | 98.52 | 76.38 | 99.68 | **100.00** | 98.19 |
| **25 samples** | Random | **100.00** ±0.0 | 96.59 ±1.0 | 93.61 ±1.5 | 98.14 ±0.3 | 90.23 ±2.9 | 99.81 ±0.3 | **100.00** ±0.0 | 99.20 ±0.4 |
| | Greedy | **100.00** | 96.74 | 93.94 | 98.27 | 83.88 | 99.71 | **100.00** | 99.41 |
| | Evo | **100.00** | 98.29 | 94.34 | 98.56 | **99.11** | **100.00** | **100.00** | 99.76 |
| | Core-set | 99.68 | 97.49 | 91.58 | 98.88 | 92.86 | **100.00** | **100.00** | 99.46 |
| | Full training | **100.00** | **99.53** | **99.20** | 98.43 | 99.08 | **100.00** | 100.00 | **99.90** |

| | Method | Pill | Screw | Tile | Toothbrush | Transistor | Wood | Zipper | Average |
|---|---|---|---|---|---|---|---|---|---|
| **1 sample** | Random | 78.71 ±5.0 | 46.22 ±4.8 | 99.59 ±0.5 | 82.58 ±3.5 | 83.31 ±4.7 | 98.18 ±0.7 | 94.58 ±1.5 | 83.61 ±1.2 |
| | Greedy | 89.23 | 55.87 | **100.00** | 88.61 | 91.42 | 99.21 | 99.37 | 89.70 |
| | Evo | 72.64 | 52.96 | 99.13 | 90.00 | 92.12 | 99.04 | 96.61 | 85.55 |
| | Core-set | 74.58 | 38.29 | 98.63 | 85.83 | 82.79 | 99.21 | 95.64 | 82.80 |
| **5 samples** | Random | 89.48 ±2.0 | 52.86 ±5.1 | 99.87 ±0.1 | 87.22 ±5.2 | 93.77 ±1.7 | 98.57 ±0.4 | 97.55 ±1.5 | 90.43 ±0.9 |
| | Greedy | 89.77 | 53.40 | **100.00** | 87.22 | 94.46 | 99.30 | 99.16 | 91.52 |
| | Evo | 91.41 | 61.47 | 99.49 | 98.33 | 97.58 | 99.30 | 98.90 | 94.12 |
| | Core-set | 85.00 | 52.82 | 99.57 | 98.89 | 96.50 | 98.77 | 95.93 | 92.13 |
| **10 samples** | Random | 90.95 ±1.8 | 58.42 ±4.0 | 99.89 ±0.1 | 90.75 ±1.1 | 95.86 ±1.8 | 98.60 ±0.2 | 98.13 ±1.0 | 92.80 ±0.6 |
| | Greedy | 93.43 | 53.68 | **100.00** | 85.56 | 96.38 | 99.30 | **99.58** | 92.94 |
| | Evo | 93.40 | 75.90 | 99.71 | 99.44 | 99.58 | 99.39 | 98.58 | 96.37 |
| | Core-set | 89.20 | 61.26 | 98.85 | **100.00** | 99.17 | 98.51 | 96.77 | 95.04 |
| **25 samples** | Random | 93.73 ±1.1 | 72.89 ±4.0 | 99.94 ±0.1 | 90.06 ±0.6 | 97.88 ±0.7 | 98.62 ±0.2 | 98.64 ±0.6 | 95.29 ±0.3 |
| | Greedy | 94.03 | 54.44 | **100.00** | 85.56 | 98.17 | 99.21 | 99.55 | 93.53 |
| | Evo | 95.23 | 95.43 | 98.77 | 99.72 | 99.50 | **99.56** | 99.55 | **98.52** |
| | Core-set | 94.38 | 89.44 | 99.57 | **100.00** | 99.58 | 98.68 | 98.37 | 96.80 |
| | Full training | **96.21** | **97.13** | 99.96 | 90.28 | **99.62** | 98.77 | 99.11 | 98.48 |

