# OpenReview forum: "How Low Can You Go? Identifying Prototypical In-Distribution Samples for Unsupervised Anomaly Detection"
_TMLR — Rejected by TMLR_

### Review · Reviewer_Uiyb · 2024-08-01

**Summary Of Contributions:**

The authors propose a that training unsupervised models like variational autoencoders or reverse distillation for anomaly detection can be done by surprisingly few samples. If these samples are a well-chosen core-set, then very few amples can even be better than training on full datasets. Finding such a core-set is part of the contribution.

**Audience:**

No

**Broader Impact Concerns:**

No concerns.

**Claims And Evidence:**

No

**Requested Changes:**

Introduction:

- "We further demonstrate that the prototypical samples identified by our method and their characteristics
translate to equally good performance for other models, datasets, and even tasks." This seems very vague.
- "Lastly, we provide a theoretical justification explaining the increase in performance through training
with very few samples." Is is possible to give a vague idea of how you are doing this (or even where in the paper you are doing this) in the introduction?

Section 2

- "Anomaly detection is deeply rooted in computer vision". I would disagree here. While there are certainly many applications of anomaly detection in CV, there are certainly similarly many anomaly detection application on time series, not to mention many other (more singular) areas.
- While Section 2.1 is called "Anomaly detection", it only deals with "Unsupervised anomaly detection in computer vision using deep learning". I guess the section name should be changed.

Section 3

- The statement "Without loss of generality, a typical neural network used for anomaly detection can be divided into two parts:" about factorizing a neural network is obviously true. Nevertheless, this factorization is very far from unique, and it is very unclear what the latent should code be in any neural network. Does it matter which latent code I choose? Did you do experiments on this?
- I know what you want to say, but this is mathematical gibberisch: (y = −1, ∀x in X_train)
- What is "the detection error"? Why use the definite article, if the term is not previously introduced.
- In formua (4), the subscript unter argmin literally say that you are taking the (fixed!) first M examples from the training set. This is probably not what you mean.
- While I know what it is, AUROC is neither introduced, nor cited, nor explained. While this is not an obscure notation, it is far from being widespread enough to not being introduced.
- "Note that the set of samples that produce the smallest errors is not necessarily equal to the set of samples that together produce the lowest error." Are you sure? I might be confused here because of the previous point. I might even be more confused because this sentence does not clearly state what it means. I might be more confused (ok, I am not), because I do not know what the detection error is.
- In Algorithm 1, what is "the best subset"?
- Below algorithm 1: "merged" how?

Section 5:

- Table 1: Which "core-set selection"?
- Table 1: The confidences $\pm a$: is this $1\sigma$? $2\sigma$? Or what?
- Table 1: Why are confidences only given for the random approach? The other approaches still have very random parts.
- Table 1: How many repetitions did you do?

Specific problems about complexity theory:

- Why is this so? "However, the problem is NP-hard." If the following sentence "For a subset size of M, there exist [...]" is supposed to be the reason, then this reason is false. I find it particularly hard to believe that it is obvious that such a problem is NP-hard, because the problem is not formalized. (Depending on the formalization, it might be NP-hard or not.) I might be confused here, because I do not know what the detection error is.
- In particular, if a paper starts approximating NP-hard problems (in this case probably even NP-complete), I would expect proofs about complexity classes like APX, FPTAS, PTAS, theoretical approximation quality, empirical approximation quality or something. I am totally lost here. (I might let this slide, if the (perhaps incorrect) mentioning of NP-hardness would not have been here. But the authors insist on being formal in complexity theory, so they should back up such statements.)
- "Since this problem is also NP-hard, we approximate the solution using an evolutionary algorithm:" I repeat my statements from above for NP-hard problems.

**Strengths And Weaknesses:**

Strengths
- If I would understand the approach (which I do not), the results and the idea seem impressive.

Weaknesses
- The formal description in this paper is very vague. Many parts are not formally described. I will put a long list of such problems under "Requested changed". Note that even if these things are fixed, I guess more problems will surface. And I do not claim that this (long) list of problems is complete.
  - Let me pick a very important point here as an example: how do you train a (deep NN) model on one sample? ("Since it is possible to train UAD models with only one sample")? Very much of your approach depends on the details of this.
  - This (and many other instances of) vagueness prevents me from evaluating the correctness of the paper.
- How do you select a core-set in you approach, without looking at the full dataset? In that sense, you are still using the full dataset. Hence, I think you need another advantage.
  - Is your approach better? It seems not, or only on some datasets.
  - Is your approach fast? I cannot tell.
- Regarding the usage of complexity theory, I also have several problems, see below.
- In general, I do not get an idea about computation times. Several subalgorithms might be very expensive. (I cannot tell, because I do not understand.)
  - Specific example: "by training the anomaly detection model on $x_i$ only and running inference on $x_k$." This seems very expensive in terms of
 computation. I might be confused here, because I do not know how you train a model on a single datapoint.
- In the experiments, the results with few datapoints are (usually and expectedly) worse than training on the full dataset. I do not understand how much worse another "stupid baseline" would be. (I know that the following analogy is not correct with AUROC-scores; it is meant in the sense that stupid baselines are needed when you claim not to be much worse.) On MNIST I could use the stupid anomaly detection method that always says "anomaly" and I would be correct in 90% of the cases. For one sample, three of the 4 approaches are below 90%.
- Table 1: Are any of the improvements statistically significant?

This paper needs a complete rewrite from scratch! It seems interesting (as far as I can tell), so I would hope a rewrite is possible and the results stand up to scrutinity then. But in the current state, the paper is very far from being publishable.

After section 5.1 I stopped reading. Some of my confusion might be clarified afterwards, but certainly new confusion would have arisen. And my review was long enough.

---

> ### Author Response · Authors · 2025-03-04
> **Rebuttal: Authors' Response to Reviewer Uiyb's Feedback**
>
> We sincerely thank the reviewer for their detailed feedback and valuable insights. We have carefully addressed each concern and hope that our responses sufficiently clarify any misunderstandings and resolve any open questions.
>
> As mentioned in our general response to all reviewers, we have significantly revised the manuscript to better communicate the core intent of our work. Based on the reviewer’s comments, we believe that certain aspects of our approach may not have been conveyed clearly in the initial version. To address this, we have refined our explanations, improved the structure of key sections, and added further clarifications where needed.
>
> Furthermore, we noticed that the reviewer has not finished reading the paper, explicitly stating that they did not continue after Section 5.1. We believe that some of the raised concerns would have been naturally answered in later sections or were already covered in previous parts of the manuscript. A full read would likely have provided additional context for many of these points. However, we also acknowledge that some of this confusion may be due to our writing, and we have worked to improve the clarity of our explanations in the revised manuscript.
>
> Given these considerations, we kindly ask the reviewer to fully re-read the revised manuscript to ensure that their concerns have been addressed accordingly and that no more confusion arises. We genuinely appreciate the time and effort spent on this review and hope that our revisions now clearly convey the novelty and significance of our contributions.
>
> 1. **Regarding "While Section 2.1 is called "Anomaly detection", it only deals with "Unsupervised anomaly detection in computer vision using deep learning". I guess the section name should be changed."**: Thank you for pointing out this ambiguity. We have modified the name of Section 2.1 to "Unsupervised Anomaly Detection" to more accurately reflect its content and ensured we are consistent in the use of the term throughout the entire manuscript.
>
> 2. **We appreciate the reviewer’s feedback regarding the clarity of our mathematical notation.**
>     - **Regarding "the notation in Section 3"**: We acknowledge that the previous formulation y = −1, ∀x ∈ X_train was unclear. We have revised this definition to explicitly state that we are learning a decision function where x is mapped to the binary labels. This revision ensures a clearer and more precise description of the labelling process.
>     - **Regarding "Formula (4)"**: We also recognize that the subscript under the argmin may have implied an unintended fixed ordering of selected samples. We have updated this formulation to clearly indicate that we are selecting an unordered subset of X_train, avoiding any potential confusion.  Overall, we have improved, refined, and extended our notation throughout the manuscript, particularly in Section 3, to ensure consistency and clarity while avoiding ambiguities. We thank the reviewer for highlighting these issues.
>
> 4. **Regarding "What is "the detection error"? Why use the definite article, if the term is not previously introduced."**: We appreciate the reviewer’s observation. We have adjusted the article regarding the detection error after Formula 2 in Section 3.
>
> 5. **Regarding "While I know what it is, AUROC is neither introduced, nor cited, nor explained. While this is not an obscure notation, it is far from being widespread enough to not being introduced."**: We thank the reviewer for pointing this out. In response to the feedback, we have added an explanation in Section 3.1 to explain the Area Under the Receiver Operating Characteristic Curve (AUROC) metric when it is first mentioned. Additionally, we have included relevant references to literature that provide a deeper explanation of the metric for readers who may be unfamiliar with it.
> 6. **Regarding ""Note that the set of samples that produce the smallest errors is not necessarily equal to the set of samples that together produce the lowest error." Are you sure?"**: Yes, we acknowledge that the previous statement may have been insufficient in explaining why this is the case. To address this, we have expanded the explanation at the end of Section 3.1.1 to provide greater clarity on the issue. Thank you for pointing this out.

---

> > ### Author Response · Authors · 2025-03-04
> > **Rebuttal: Authors' Response Cont.**
> >
> > 6. **Combined response regarding the concerns on Algorithm 1:**
> >     - **Regarding "what is 'the best subset'?"**: The best subset refers to the population P' after P/2 individuals have been removed from P, according to the fitness function (Formula 5), at each iteration of the evolutionary algorithm. In response to this comment, we have clarified and enhanced our notation to make this concept clearer.
> >     - **Regarding "'merged' how?"**: While the procedure for merging two individuals was outlined in Algorithm 1, we recognize that the explanation may not have been detailed enough. As such, we have revised and expanded the explanation in Section 3.1.2 to provide a more thorough description of the crossover and mutation operations.
> > 7. **Combined Response to Comments on Table 1:**
> > We appreciate the reviewer’s feedback regarding Table 1 and have taken steps to ensure greater clarity.
> >     - **Regarding "which core-set selection?" (6)**: We believe the reviewer may have misunderstood our table. The description states that the methods listed include a random baseline (completely random selection), greedy weakly supervised method, evolutionary weakly supervised method, and our unsupervised core-set selection approach. There is only one core-set selection approach, which is our proposed method. While the weakly supervised baselines could, in a broader sense, be interpreted as core-set selection methods, that is not their intended role in our work. To eliminate any potential confusion, we have further clarified this distinction in the entire manuscript and updated the caption of Table 1.
> >     - **Regarding confidence values for the random method (Comment 7)**: The numbers in Table 1 for the random method refer to the standard deviation, as now noted in the table description. Since random sampling can lead to significant performance variation depending on the selected samples, we report this variability. We have now updated the caption of Table 1 to explicitly clarify this point and ensure there is no room for misunderstanding.
> >     - **Regarding why we only provided the standard deviation for the random baseline (8) & Regarding the number of repetitions (9)**: We provide a standard deviation only for the random baseline because our core-set selection algorithm was highly stable, consistently finding solutions within a narrow range, making additional variance reporting unnecessary. However, we confirm that all methods were evaluated over multiple iterations.
> >     For the greedy and evolutionary algorithms, which are weakly supervised, their variability primarily stems from the fact that they rely on labels and a score derived from training our UAD model on each individual sample. The main source of randomness here would come from the training process itself. However, since each sample is trained on individually, which is computationally costly, we did not repeat this process extensively. The inherent randomness in training is however significantly reduced, particularly when dealing with only a single sample.
> >     While the evolutionary algorithm does incorporate randomness in its mutation, replacement strategy, and crossover mechanisms, as mentioned in Section 4.2 (Experimental Setup), we found that it was highly stable despite these variations.
> > 8. **Regarding "Let me pick a very important point here as an example: how do you train a (deep NN) model on one sample? ("Since it is possible to train UAD models with only one sample")? Very much of your approach depends on the details of this."**: We appreciate the reviewer’s concern regarding training a deep neural network on a single sample. However, the process remains fundamentally the same as training on a larger dataset. The model still undergoes standard optimization steps, with the key difference being that the batch size is one, meaning each training iteration updates the model based on just that one sample. While this setup leads to extreme overfitting, it does not break the training process itself. The concept of an epoch becomes less meaningful in this case, as a single epoch consists of only one batch containing the lone sample. Nonetheless, the gradient-based optimization, weight updates, and backpropagation function identically to training with a full dataset.

---

> > ### Author Response · Authors · 2025-03-04
> > **Rebuttal: Authors' Response Cont.**
> >
> > 9. **Regarding "How do you select a core-set in you approach, without looking at the full dataset? In that sense, you are still using the full dataset. Hence, I think you need another advantage. Is your approach better? It seems not, or only on some datasets. Is your approach fast? I cannot tell."**: We appreciate the reviewer’s question and would like to clarify the distinction between our fully unsupervised core-set selection approach and the weakly supervised baselines. Our unsupervised core-set selection approach is very fast, whereas the other two methods introduce computational overhead but are only included as baselines to benchmark our unsupervised method, not as practical recommendations. Regarding the concern that we still use the full dataset: Yes, we do initially consider the full dataset, but the key takeaway from our work is that we can extract a much smaller subset that still achieves strong, sometimes even superior, performance compared to training on the full dataset. This directly supports the quality over quantity argument, demonstrating that some training samples may negatively impact performance, while a carefully selected subset can yield optimal results. Additionally, as highlighted in Section 1: Introduction, training with a significantly smaller dataset offers numerous advantages beyond the obvious efficiency gains. To better address this in the manuscript, we have elaborated on the limitations in the new Section 5.7: Limitations and Practical Considerations, where we discuss computational trade-offs and practical implications. We thank the reviewer for prompting us to refine this explanation.
> > 10. **Regarding "In the experiments, the results with few datapoints are (usually and expectedly) worse than training on the full dataset. I do not understand how much worse another "stupid baseline" would be."**: We appreciate the reviewer’s comment. As noted in Table 1, we have included a fully random baseline, as a "stupid" baseline. We encourage the reviewer to refer to Table 1 and observe the results, as well as the discussion of these results in Section 5.1. Even random subsets can achieve considerable performance, albeit with higher variability (larger standard deviation), depending on the samples selected. The performance is particularly strong when the random subset happens to resemble the samples selected by our core-set selection method. In regards to this comment, we have elaborated on this behavior and its implications in the manuscript.
> > 11. **Regarding "Are any of the improvements statistically significant?"**: Thank you for raising this point. While we recognize that statistical significance tests and confidence intervals are commonly used in practice, we would like to emphasize that the primary goal of our study is not to directly outperform specific benchmarks or models. Instead, our focus is on providing evaluations across various methods, models, datasets, and iterations to ensure robustness and consistency. Nevertheless, we acknowledge this as a limitation and encourage future work to further validate our findings using formal statistical tests where appropriate.
> > 12. **Regarding ""We further demonstrate that the prototypical samples identified by our method and their characteristics translate to equally good performance for other models, datasets, and even tasks." This seems very vague."**:  We thank the reviewer for their opinion. However, we feel that at this point in the manuscript, it meets the right level of abstraction. It is listed as part of our main contributions in the introduction, and we do not believe it is necessary to enumerate all the different combinations of models, datasets, and tasks explored in this manuscript at that point. The introduction is meant to provide a high-level summary rather than exhaustive details. However, to ensure clarity for all readers, we have now included a direct reference to the section where this is discussed in detail, allowing readers to easily locate and verify the specific experiments and findings. We appreciate the reviewer’s feedback.
> > 13. **Regarding "Lastly, we provide a theoretical justification explaining the increase in performance through training with very few samples." Is is possible to give a vague idea of how you are doing this (or even where in the paper you are doing this) in the introduction?"**: We thank the reviewer for pointing this out. We acknowledge that referring to this as a "theoretical justification" may have been somewhat misleading, and we have revised the wording accordingly. What we intended to convey is that, based on the long-tail hypothesis, most in-distribution samples exhibit low inter-sample variance. Therefore in Section 5.4, we analyze the tails of our distributions in feature space and show that the samples in the long tails tend to be poor-quality, and are not selected by our core-set selection method. We have updated our wording to more accurately reflect this explanation

---

> > ### Author Response · Authors · 2025-03-04
> > **Rebuttal: Authors' Response Cont.**
> >
> > 14. **Regarding "Anomaly detection is deeply rooted in computer vision". I would disagree here. While there are certainly many applications of anomaly detection in CV, there are certainly similarly many anomaly detection application on time series, not to mention many other (more singular) areas."**: We acknowledge the reviewer’s point and clarify that it was not our intention to claim that unsupervised anomaly detection originates solely from computer vision. We recognize that anomaly detection has strong(er) foundations in multiple domains, including time series analysis and other fields. To ensure clarity and broader accuracy, we have revised the statement to say that "anomaly detection is deeply rooted in many domains, including computer vision." We believe this more accurately reflects the widespread applicability of anomaly detection across various disciplines.
> >
> > 15. **Regarding "The statement "Without loss of generality, a typical neural network used for anomaly detection can be divided into two parts:" about factorizing a neural network is obviously true. Nevertheless, this factorization is very far from unique"**: We agree with the reviewer that our original wording may have been too general. To address this, we have refined our formulation to specifically refer to the models used in this paper (see the second paragraph of Section 3), which are structured in this manner. Additionally, we have improved our notation and explanation to more clearly introduce the concept of factorizing a neural network into a feature extractor and a predictor, ensuring that it is properly contextualized within the scope of our work. We appreciate the reviewer’s feedback and have made these clarifications accordingly.
> >
> > 16. **Regarding "The concerns raised about complexity theory"**: We thank the reviewer for their comments regarding our use of complexity theory terminology. We acknowledge that stating a problem is NP-hard without providing formal proofs or explicit references to well-established results in the literature is not best practice. While our original intention was to emphasize the computational difficulty of selecting an optimal subset, we recognize that the precise complexity classification is not central to our work. Given this, we have revised our wording in Section 3 to avoid using the term "NP-hard" and instead simply highlight that the problem is computationally challenging. We appreciate the reviewer’s input and have made these changes accordingly.

---

> > > ### Comment · Reviewer_Uiyb · 2025-03-08
> > > **Re: Rebuttal**
> > >
> > > I thank the authors for their many changes to the paper. The points in my review have been addressed. The paper is also much clearer to read in the parts I did not criticise. The current quality of the write-up is very much better than the original version.
> > >
> > > I have some minor points that could still be improved, but that are not a reason so reject the paper:
> > >  - The anomaly detection is seems still very much rooted in the usual methodology of images. It might be suitable to make that point clearer or acknowledge the vast area of anomaly detection with their differences. (This point was already a problem in the original paper, has now somewhat improved, but should still be made clearer.)
> > >  - Section 5.5 might be a strong point of the paper. This subsection might be improved by adding more examples or results. Or by discussing additional points in which cases data should "warrant special consideration in downstream task".
> > >
> > > I would be willing to suggest (a weak) acceptance of the paper, but would not champion the paper in case of doubts by the other reviewer.

---

> > > > ### Comment · Action_Editor_Huzq · 2025-03-10
> > > >
> > > > Reviewer Uiyb, Please add a comment in your "Official Recommendation" (near the top of all the reviews)

---

### Review · Reviewer_TaMH · 2024-08-19

**Summary Of Contributions:**

The paper studies the role of the size of the training datasets in the problem of anomaly detection. In particular, the paper debunks the common wisdom that more training data lead to improved performance. For the anomaly detection task, it seems that quality plays more important role than quantity. The paper shows that with a few training samples it achieves comparable and in certain cases improved performance vs. methods using the entire training set. Based on that finding, the paper proposes to use core-set selection to extract representative examples from the training sets. Experimental results support the claims in the paper.

**Audience:**

Yes

**Claims And Evidence:**

Yes

**Requested Changes:**

W1. A supervised measure is used during optimization, which is confusing

The paper focuses on unsupervised anomaly detection, yet AUCROC is used to optimize the selection. Isn't that contradictory to the initial claim, or is AUCROC used without seeing labels? I am missing something here.

W2. Missing simple baselines

How a simple baseline like k-means performs in that task. It should be fast to just run k-means on the given train set and then use only a representative example per cluster or multiple representative examples.

W3. Runtime results are missing

How much time does it take to construct the representative examples? This is like a chicken-egg problem. You need a train set first to then reduce it. In contrast, we would ideally need an approach that without supervision tell us which samples make sense to annotate/consider (but without seeing the training set). Only once you see the final results you know if this solution is OK or not. The approach does not provide any guarantees. You may try first with 5 samples, if they are not enough you need to use more. But that defies the initial purpose, the question is that we need a method to tell us initial which samples to use.

**Strengths And Weaknesses:**

Strengths:

S1. Unsupervised anomaly detection is critical for modern monitoring applications and smaller dependence on training sets is important for this task
S2. Interesting finding about the use of less carefully selected data
S3. Experimental results support the general claims

Weaknesses:

W1. A supervised measure is used during optimization, which is confusing
W2. Missing simple baselines
W3. Runtime results are missing

---

> ### Comment · Action_Editor_Huzq · 2025-02-20
> **2-week extension on rebuttal was granted by Editor-in-Chief today**
>
> 2-week extension on rebuttal was granted by Editor-in-Chief today.  Hence, we'll ask for your recommendation after March 5th.

---

> ### Author Response · Authors · 2025-03-04
> **Rebuttal: Authors' Response to Reviewer TaMH's Feedback**
>
> We would like to express our gratitude to the reviewer for their thoughtful comments on our work. We are happy that no fundamental issues were raised in the feedback. Their comments were instrumental in identifying key areas that required further clarification and discussion. We believe that we have addressed all of the reviewer’s concerns in the revised manuscript. In the following, we provide detailed responses on each of the points raised:
> 1. **Regarding "A supervised measure is used during optimization, which is confusing. The paper focuses on unsupervised anomaly detection, yet AUCROC is used to optimize the selection. Isn't that contradictory to the initial claim, or is AUCROC used without seeing labels? I am missing something here."**: To clarify, all UAD models in our study are trained in a fully unsupervised manner. Of course, we use a labelled validation set for performance evaluation, where AUROC serves as a metric for this evaluation. However, it is important to distinguish between the model training process, which remains fully unsupervised, and the sample selection algorithms, which are used to identify these subsets. The two sample selection methods (greedy and evolutionary) are weakly-supervised baselines that use the AUROC metric, but are included solely to benchmark the proposed unsupervised core-set selection technique. This proposed method for core-set selection is fully unsupervised and does not utilize AUROC (see the beginning of Section 3). In response to this feedback, we substantially revised Section 3.
> 2. **Regarding "Missing simple baselines: How a simple baseline like k-means performs in that task. It should be fast to just run k-means on the given train set and then use only a representative example per cluster or multiple representative examples."**: We sincerely appreciate the reviewer’s suggestion regarding the use of k-means as a baseline. While we understand that k-means could be a simple and effective approach for identifying high-performing subsets (e.g., by selecting cluster centroids or representative samples), the primary objective of our work is to demonstrate the phenomenon that UAD models can achieve competitive performance with significantly reduced training data. Our experiments already include a simple random selection baseline, which serves to verify this initial observation. Therefore, benchmarking multiple algorithms, including k-means, is outside the scope of this paper as would divert from the main focus, which is to showcase, investigate and discuss the phenomenon itself. However, we believe that such comparisons should be extended in future research. In response to this comment, we have added "Section 6: Future Work" and specifically refer die reviewer to "Future work should focus on refining our core-set selection algorithm to enhance its ability to extract the most representative and informative training samples while also exploring alternative core-set selection methods that may provide even better results." for more details.

---

> > ### Author Response · Authors · 2025-03-04
> > **Rebuttal: Authors' Response Cont.**
> >
> > 3. **Regarding "How much time does it take to construct the representative examples? This is like a chicken-egg problem. You need a train set first to then reduce it. In contrast, we would ideally need an approach that without supervision tell us which samples make sense to annotate/consider (but without seeing the training set). Only once you see the final results you know if this solution is OK or not. The approach does not provide any guarantees. You may try first with 5 samples, if they are not enough you need to use more. But that defies the initial purpose, the question is that we need a method to tell us initial which samples to use."**: We appreciate the reviewer’s perspective on the computational cost and practicality of subset selection. It is indeed true that identifying optimal subsets requires an initial training set and iterative evaluation, which can be computationally demanding. However, our proposed unsupervised core-set selection technique is designed to be very efficient in identifying a set of prototypical samples. Therefore, we believe the reviewer may be referencing the weakly supervised baselines mentioned in our work, which, as pointed out, are less efficient and computationally expensive. However, these are only baselines, meant to illustrate our point and we do not suggest using these methods in practice. To clarify, and in response to this comment, we have added Section 5.7 (Limitations and Practical Considerations) to our manuscript. Please refer to the line "A key practical limitation of these baselines is their computational cost..." for more details. Finally, while an initial training dataset is necessary to identify a smaller subset, our results indicate that a larger training set may contain samples that actually harm model performance. Therefore, the core-set selection approach could help filter out these less informative samples, thereby improving the overall quality of the dataset. Our findings suggest that a small number of high-quality samples can yield strong results, which is central to the key takeaway of our paper.

---

### Review · Reviewer_5k1P · 2025-01-09

**Summary Of Contributions:**

Presents and evaluates basic coreset selection techniques for anomaly detection in the vision domain.

All methods make use of a suitable backbone model to obtain latent representations, and then select a small subset of the training examples for training the final anomaly detection model. One technique is unsupervised (based on a GMM), the remaining techniques use validation data, either based using independent selection ("greedy") or set selection ("evolutionary").

The experimental study compares the approach against certain baseline models trained on full data.

**Audience:**

Yes

**Claims And Evidence:**

No

**Requested Changes:**

-

**Strengths And Weaknesses:**

S1. Very simple methods.

S2. Provides further evidence that, in some cases, only a few well-chosen examples can give good results for anomaly detection.

W1. Related work not adequately represented. First, prior methods for coreset selection are only briefly mentioned but not related to the presented methods (which are, in fact, straightforward). E.g., what about simply using methods for coreset selection for clustering? Second, the base models used in the experimental study do not actually represent the SOTA. For example, the authors used PatchCore for the MVTec-AD dataset and report a full data AUROC of 98.48. The original PatchCore paper reports 99.1, however, and paper-with-code lists multiple methods with higher scores (up to 99.9). Third, what about in-context learning, which also uses few-shot prompts? E.g., Zhu and Pang, "Generalist Anomaly Detection via In-context Residual Learning with Few-shot Sample Prompts", CVPR24.

W2. Experimental study hard to interpret. (i) It's not clear how hyperparameters have been selected (and what they are), how data for the weakly supervised models have been obtained, and how the actual experiments were run. (ii) Statistical significance is unclear, no confidence intervals are given. (iii) Results are mixed, with little discussion on why.

W3. Greedy method not convincing. First, the proposed method is not actually greedy but performs independent selection. A greedy method would select a the best single example first, then add the best second one, then the best third one, etc. It's likely that this actual greedy approach outperforms the presented approach (and perhaps also the evolutionary approach).

W4. Discussion lacks nuance. (i) It's clear that an approach that describes in-distribution data by only a few examples is inherently limited to certain scenarios; such a discussion is completely lacking. (ii) The paper focuses on vision with strong backbone models, but is presented much more generally. I find this misleading. (iii) It's not discussed why one would use few examples in the first place. In fact, in most cases, using all available data was better or close in the experimental study. (iv) I do not find the discussion around long-tail convincing. Anomalies are rare (in contrast to what's illustrated in Fig. 6) but the long tail can be heavy.

---

> ### Author Response · Authors · 2025-03-04
> **Rebuttal: Authors' Response to Reviewer 5k1P's Feedback**
>
> We would like to thank the reviewer for their time and effort in reviewing our paper. Their feedback has highlighted areas where the content and clarity of our writing could be improved, and we appreciate the opportunity to refine these sections. At the same time, we recognize that some of the concerns raised may have resulted from ambiguities or unintended misinterpretations in our original writing. Given the importance of these aspects to the paper’s overall message, we have put an emphasis on clarifying these points and thus kindly ask the reviewer to reevaluate the revised manuscript.
>
> In the following we provide detailed responses to each point that was raised in the feedback:
>
> 1. **Regarding "It's not clear how hyperparameters have been selected (and what they are)"**: In response to the reviewer’s concern, we have significantly revised and expanded Section 4.2 to further clarify the selection and rationale behind the chosen hyperparameters, ensuring that the details are transparent and easily understandable. The definition and explanation of all hyperparameters are provided in Section 3, where we introduce both our unsupervised core-set selection strategy and the weakly supervised baselines. Additionally, the specific hyperparameters used for the configurations of these algorithms are detailed in Section 4.2 (Experimental Setup).
> 2. **Regarding "How data for the weakly supervised models have been obtained":** We thank the reviewer for pointing this out. In response to this comment we have revised Section 4.2 (Datasets and Models) and Appendix A, where now all relevant details regarding the datasets used in this study are provided. The data used for the weakly supervised baseline algorithms is the same as that employed for all other methods and models in the paper and is publicly available. The key difference lies in how the data is utilized: we first trained our UAD models on each sample in the datasets individually, recorded the resulting scores on the validation set, and then used these scores to build the subsets, as described in Section 3.1.1 ("To achieve this, we train our model..."). In response to this, we have thoroughly revised the whole of Section 3 in order to address and clarify all related questions.
> 3. **Regarding "Signifiance and confidence intervals"**: Thanks for bringing this up. While we agree that statistical significance tests and confidence intervals are standard practice, we would like to remark that the goal of our study is not to outperform specific benchmarks or models. Instead, our findings are based on evaluations across multiple methods, models, datasets, and iterations, ensuring robustness and consistency. However, we acknowledge this as a limitation and encourage future work to further validate our findings using formal statistical tests where appropriate.
> 4. **Regarding "Related work not adequately represented. First, prior methods for coreset selection are only briefly mentioned but not related to the presented methods (which are, in fact, straightforward). E.g., what about simply using methods for coreset selection for clustering?"**: The primary focus of our paper is not on core-set selection itself, but rather on demonstrating the phenomenon of achieving effective anomaly detection with a small number of carefully selected samples, for which we happened to utilize core-set selection methods. As such they were included only as a tool to validate our core observation. The intention was not to propose the best core-set selection strategy for this specific use case but to illustrate the broader concept. However, we view the use of clustering methods as a potential direction for future work, which we have highlighted more generally in the revised version of the paper. Therefore, and in response to this feedback, we have extended the manuscript to include Section 6 (Future Work).
> 5. **Regarding "The base models used in the experimental study do not actually represent the SOTA. For example, the authors used PatchCore for the MVTec-AD dataset and report a full data AUROC of 98.48. The original PatchCore paper reports 99.1, however, and paper-with-code lists multiple methods with higher scores (up to 99.9)."**: The base models used in the experimental study were state-of-the-art (SOTA) at the time of writing the paper. We acknowledge that since then, newer methods have emerged, and some of them have surpassed the results reported in our study. However, we believe it is important to consider the models as SOTA at the time of experimentation, given that the review process took longer than anticipated. Additionally, we have no indication that our observations do not generalise for the lastest approaches.

---

> > ### Author Response · Authors · 2025-03-04
> > **Rebuttal: Authors' Response Cont.**
> >
> > 6.  **Regarding "What about in-context learning, which also uses few-shot prompts? E.g., Zhu and Pang, "Generalist Anomaly Detection via In-context Residual Learning with Few-shot Sample Prompts", CVPR24."**: The mentioned work from CVPR24 was published shortly before our submission deadline. As a result, we were unable to incorporate references to newly published material such as CVPR24 in our manuscript. That being said, the approach presented in their work is certainly intriguing, and it would be interesting to explore whether our observations could be extended to this scenario in future research. Therefore, in Section 6 on Future Work, we have included a statement ("Additionally, it is crucial to extend this investigation...") emphasizing the need to explore additional datasets and models to fully investigate the generalizability of the observed phenomenon.
> > 7. **Regarding "Greedy method not convincing. First, the proposed method is not actually greedy but performs independent selection. A greedy method would select a the best single example first, then add the best second one, then the best third one, etc. It's likely that this actual greedy approach outperforms the presented approach (and perhaps also the evolutionary approach)."**: We appreciate your concern about the greedy method. However, we respectfully disagree that our approach does not align with the greedy paradigm. A greedy algorithm does not necessarily require sequential dependency between steps—it simply makes locally optimal choices (given the chosen heuristic) at each decision point. Our method selects the top n samples based on their individual performance, following the greedy principle of always choosing the best available option. That being said, we acknowledge that a stepwise greedy selection strategy (where samples are added iteratively based on joint performance) could potentially yield better performing subsets. At the same time, such an approach would also introduce substantially higher computational costs. Nonetheless, exploring this trade-off in future work would be an interesting direction.
> > 8. **Regarding "It's clear that an approach that describes in-distribution data by only a few examples is inherently limited to certain scenarios; such a discussion is completely lacking."**: We agree that reducing the training dataset to just a few examples, and thereby limiting the representation of in-distribution data, may have inherent limitations. Furthermore, there may be ethical and fairness concerns associated with such a method, as it could potentially result in a biased representation. In response to this concern, we have expanded the discussion with a new Section 5.7 (Limitations and Practical Considerations) to address these potential limitations. However, and specifically regarding the use-cases in our work, as long as the test performance after training on a reduced dataset is comparable to or better than when using the full training set, the key information necessary for the task seems to remain in the selected subsets. Nevertheless, we agree that this observation requires further investigation, particularly across different datasets and tasks, to fully understand its generalizability and limitations.
> > 9. **Regarding "How the actual experiments were run"**: We again thank the reviewer for pointing out that there is some confusion. In response to this and the previous comment, we have updated and expanded Section 4.2 to provide clearer, more detailed information on the experimental procedure. As mentioned previously, the details of how the experiments were conducted are outlined in Section 4.2 (Experimental Setup), which describes the experimental setup and the configuration of the algorithms.
> > 10. **Regarding "Results are mixed, with little discussion on why"**: We agree there is a variability in the performance of the subsets (e.g., 25 samples). However, it is important to note that the key takeaway from our results is consistent: training with a small subset of carefully selected samples yields performance that is close to that achieved using the full dataset. The observed differences in performance between algorithms were not intended to be the focus of our study, nor were we aiming to identify the best performing method for every subset size or dataset combination. Once again we highlight that, our goal was to demonstrate that even small subsets of high-quality samples can deliver comparable results to full training. However, we acknowledge that the variation in algorithmic performance provides an interesting direction for future work, where further refinement could yield more consistent or optimized results.

---

> > ### Author Response · Authors · 2025-03-04
> > **Rebuttal: Authors' Response Cont.**
> >
> > 11. **Regarding "The paper focuses on vision with strong backbone models, but is presented much more generally. I find this misleading."**: We respectfully disagree with the reviewer’s assessment. The results are clearly communicated with respect to the state-of-the-art models available at the time of the study, specifically within the context of vision tasks. Given that our work explores unsupervised anomaly detection, it is natural to utilize strong backbone models, as they are the most effective in showcasing the phenomenon we aim to highlight. However, we do not claim that the observation is exclusive to these models. In fact, the version of the FAE model used in our experiments is already a slightly scaled-down version compared to its original form, which suggests that our findings are not dependent on high-performance models alone. We believe our observations can extend beyond strong backbones and are likely applicable to a range of models, even those with lower capacity.
> > 12. **Regarding "It's not discussed why one would use few examples in the first place. In fact, in most cases, using all available data was better or close in the experimental study."**: It is true that the selected subsets did not always outperform or match the performance of the full dataset in our experiments, yet this does not diminish the core observation we aim to convey. The primary purpose of our study is to demonstrate that it is possible to achieve competitive results with a surprisingly small fraction of the original dataset. This highlights an important insight: curating a smaller, higher-quality dataset could be more beneficial than simply using all available data. We are not suggesting that our core-set selection approach should replace existing methods for training on large datasets. Rather, the goal is to emphasize the potential of selecting higher-quality, smaller subsets from larger datasets. We believe that future work in developing methods for curating such datasets could offer significant value as now stated in Section 6 (Future Work).
> > 13. **Regarding "I do not find the discussion around long-tail convincing. Anomalies are rare (in contrast to what's illustrated in Fig. 6) but the long tail can be heavy".**:  We thank the reviewer for their assessment, however, we respectfully disagree with the interpretation of our discussion. As stated in the caption of Fig. 6, the figure is intended solely as an illustration of our hypothesis. Furthermore, it is important to differentiate between the long tail of the in-distribution and anomalies from the out-distribution. While anomalies are rare and typically belong to the out-distribution, the long tail is part of the in-distribution. There does not necessarily have to be a direct relationship between these two. Our hypothesis is that samples from the long tail of the in-distribution may, in some cases, actually hinder the training process rather than contribute positively. This observation is independent of the anomalies in the out-distribution, as they are not part of the training dataset and do not affect the specific point we are making about data quality within the training set.

---

> > > ### Comment · Reviewer_5k1P · 2025-03-18
> > > **Thoughts on revision**
> > >
> > > I'd like to thank the authors for their detailed responses.
> > >
> > > Although some points are now clarified, the key points---i.e., the weak points of my original review---are still unaddressed. My impression of the paper remains essentially unchanged. I focus on some of these points here.
> > >
> > > On W1: Related work not adequately represented.
> > >
> > > > We show for the first time that an exceedingly small number of training
> > > > samples can suffice for performant, robust, and interpretable UAD
> > >
> > > The PatchCore paper (2022) also used 1-50 training examples and showed this. This invalidates many of the arguments given in the response.
> > >
> > > > The base models used in the experimental study were state-of-the-art (SOTA) at
> > > > the time of writing the paper. We acknowledge that since then, newer methods
> > > > have emerged
> > >
> > > I argued that PatchCore already had better numbers. This response clearly does not apply as PatchCore is from 2022 and it's used in this paper as a base model.
> > >
> > > > This challenges the common assumption that larger training datasets always
> > > > lead to better model performance.
> > >
> > > This is not a common assumption.
> > >
> > > > Our findings indicate that the composition of the training set is may more
> > > > important than its sheer size.
> > >
> > > Likewise.
> > >
> > > > Core-Set Selection Methods [...] were used purely as tools [...] The intention
> > > > was not to propose the best core-set selection strategy
> > >
> > > Then the paper shouldn't claim these methods (which it still does) and instead use prior methods. E.g., PatchCore uses a "real" greedy approach, which can directly be used on feature vectors.
> > >
> > > On W2: Experimental study hard to interpret.
> > >
> > > > Significance and confidence intervals missing
> > >
> > > Still not given; it's not clear which results are actually significant.
> > >
> > > > Results are mixed, with little discussion on why
> > >
> > > Still the case.
> > >
> > >
> > > On W3: Greedy method not convincing.
> > >
> > > > We appreciate your concern about the greedy method.
> > >
> > > Let's not argue too much about point vs subset selection. The greedy method I was suggesting is straightforward and is reminiscent to the one used in PatchCore, for example.
> > >
> > > On W4: Discussion lacks nuance.
> > >
> > > > It's clear that an approach that describes in-distribution data by only a few
> > > > examples is inherently limited to certain scenarios; such a discussion is
> > > > completely lacking.
> > >
> > > The revision now has some discussion around this, but I do not find it very satisfactory. I do agree that the coreset approach is suitable for certain distributions, but the paper does not investigate for which ones. It's probably easy to construct a synthetic distribution where the coreset approach fails (e.g., n "normal" and n "abnormal" clusters scattered randomly -> n examples needed).
> > >
> > > This also relates to my argument around long-tail distributions, which I still have.
> > >
> > > Finally, the paper is still very much vision-focused with strong backbones.

---

> > > > ### Comment · Action_Editor_Huzq · 2025-03-18
> > > >
> > > > Reviewer 5k1P, Thanks for your comments.  Please update your "official recommendation" near the top of the web page.

---

### Author Response · Authors · 2025-01-13
**Request for extension**

Dear Editor and Reviewers,

We are happy and grateful to have received three reviews for our paper and would like to address them appropriately. Unfortunately, a substantial part of our team is not available this and next week and we would thus kindly request an extension to our deadline so we can provide in-depth responses to all comments and suggestions.

Best,
the Authors

---

> ### Comment · Action_Editor_Huzq · 2025-01-31
>
> Editor-in-chief: would you like to grant an extension?

---

> > ### Comment · Action_Editor_Huzq · 2025-02-20
> > **2-week extension on rebuttal was granted by Editor-in-Chief today.**
> >
> > 2-week extension on rebuttal was granted by Editor-in-Chief today. Hence, finish your rebuttal by March 5th

---

### Author Response · Authors · 2025-03-04
**Rebuttal: General Response to All Reviewers**

We would like to sincerely thank all reviewers for their detailed feedback, constructive criticism, and insightful comments. Your input has been invaluable in helping us refine and improve our manuscript. In response to your concerns, we have undertaken a substantial revision of the paper, rewriting almost every section to improve clarity, precision, and mathematical rigour. This includes enhancing our explanations, refining our mathematical notation, reformatting key aspects, and, most importantly, ensuring that the core message of our work is more clearly communicated.

**Clarifying the Intent of Our Paper**

We recognize that our paper may not have clearly conveyed its primary objective, which is not to develop new core-set selection methods but to investigate the surprising phenomenon of effective learning from minimal subsets. The main contribution of our work is to demonstrate that a deep UAD model can be trained on a very small subset of the training dataset and still achieve performance comparable to, or even exceeding, models trained on the full dataset.
This challenges the common assumption that larger training datasets always lead to better model performance and we believe this insight is critical, as it suggests that dataset size may not be as crucial as previously thought. Our findings indicate that the composition of the training set is may more important than its sheer size.
To make the importance of this phenomenon more explicit, we have significantly revised the Introduction to elaborate on why smaller training datasets could be beneficial. Additionally, our revised manuscript now emphasizes that our focus is on investigating and explaining this phenomenon rather than proposing the best possible subset selection algorithm.

**Clarifications on Core-Set Selection Methods**

To investigate this phenomenon, we employed different subset selection approaches—one unsupervised and two weakly supervised methods. However, these methods were used purely as tools to obtain training subsets where the phenomenon could be observed. Our primary goal was not to benchmark core-set selection techniques but to highlight the surprising effectiveness of training on minimal subsets.

To ensure clarity, we have explicitly revised the manuscript to emphasize that:
- Our unsupervised core-set selection method serves as a promising approach for identifying high-performing subsets, but it is not yet an optimized solution. We believe it has potential and encourage future research to refine and improve it.
- The weakly supervised methods were included solely as baselines for comparison. Due to their reliance on labeled data and computational inefficiencies, we do not suggest their use in practical scenarios.
- In the newly introduced Future Work section, we advocate for further development of better unsupervised core-set selection techniques to fully harness the potential benefits of this phenomenon.

**Encouraging Reviewers to Re-Evaluate the Manuscript**

Given the depth of revisions, we would like to encourage all reviewers to reevaluate the updated manuscript. We hope that with these changes, our core message will be communicated more effectively and that the significance of our findings will be clearer. We believe that understanding this phenomenon is important because it may provide valuable insights into UAD training. In particular, our findings suggest that it could challenge existing assumptions about dataset size and its role in model performance, potentially opening new avenues for more efficient and effective training strategies.

Once again, we deeply appreciate the time and effort the reviewers have invested in providing their feedback. Your comments have significantly improved the quality and clarity of our work, and we thank you for contributing to this process.

---

### Decision · Action_Editor_Huzq · 2025-03-20

**Recommendation:** Reject

**Comment:**

After rebuttal and revision, the paper has improved to generally satisfy one reviewer, but he would not champion the paper.  The other two reviewers still have significant concerns and are negative on the paper.

The paper lacks comparison with more recent methods (e.g. PatchCore) to support their contributions.   Also, results are mixed and further discussion on the reasons would be necessary.   Adding runtime results to demonstrate efficiency would strengthen the contributions.

**Audience:**

Those who are interested subset selection for anomaly detection might find the paper interesting.

**Claims And Evidence:**

The authors propose selecting subsets of prototypical samples for anomaly detection.   For subset selection, they use GMM for finding core-sets (Core-set).   They also discuss a greedy algorithm and an evolutionary algorithm (Evo) for comparison.

Using 7 image datasets, they compare the different algorithms with different subset sizes.  They found that Evo in 3 datasets and Core-set in 2 datasets, selecting 25 samples could outperform the method with all data.    They further analyze the selected samples versus the discarded ones and identify characteristics in the selected samples.